# SCALING SENTENCE EMBEDDINGS WITH LARGE LANGUAGE MODELS

## ABSTRACT

Large language models (LLMs) have recently garnered significant interest. With in-context learning, LLMs achieve impressive results in various natural language tasks. However, the application of LLMs to sentence embeddings remains an area of ongoing research. In this work, we propose an in-context learning-based method aimed at improving sentence embeddings performance. Our approach involves adapting the previous prompt-based representation method for autoregressive models, constructing a demonstration set that enables LLMs to perform in-context learning, and scaling up the LLMs to different model sizes. Through extensive experiments, in-context learning enables LLMs to generate high-quality sentence embeddings without any fine-tuning. It helps LLMs achieve performance comparable to current contrastive learning methods. By scaling model size, we find scaling to more than tens of billion parameters harms the performance on semantic textual similarity (STS) tasks. However, the largest model outperforms other counterparts and achieves the new state-of-the-art result on transfer tasks. We also fine-tune LLMs with current contrastive learning approach, and the 2.7B OPT model, incorporating our prompt-based method, surpasses the performance of 4.8B ST5, achieving the new state-of-the-art results on STS tasks.

## 1 INTRODUCTION

Sentence embeddings is a fundamental problem in natural language processing, requiring language models to project sentences into a vector space based on their semantics. Current methods based on contrastive learning, such as SimCSE Gao et al. (2021), have successfully leveraged pretrained language models to generate high-quality embeddings. A significant amount of research has been devoted to refining the contrastive learning framework in order to further improve sentence embeddings Chuang et al. (2022); Wu et al. (2022a;b); Cheng et al. (2023).

Recently, large language models (LLMs), such as GPT-3 Brown et al. (2020) and LLaMA Touvron et al. (2023a), have demonstrated significant potential on various natural language processing tasks such as translation, question answering, and text classification. Current research has also explored the application of LLMs for data augmentation in sentence embeddings. By generating better sentence pairs for contrastive learning, LLMs can help alleviate the scarcity of labeled data Cheng et al. (2023); Zhang et al. (2023). However, directly utilizing LLMs to generate sentence embeddings presents two primary challenges. Firstly, LLMs, as autoregressive models, produce text instead of vectors, which necessitates vectorizing the output. Secondly, it is crucial to determine an effective approach for incorporating the capabilities of in-context learning into sentence embeddings.

In this work, we aim to investigate the capabilities of current LLMs for sentence embeddings, facilitated by the availability of open-source LLMs Touvron et al. (2023a); Zhang et al. (2022). We address the following research questions: 1) How can LLMs be used to represent sentence embeddings, and does prompt engineering, as demonstrated by PromptBERT Jiang et al. (2022) help? 2) Can in-context learning Liu et al. (2023) enhance the quality of sentence embeddings? 3) Does the scaling up the model parameters stil work when the number of parameters exceeds billions? 4) What improvements can be achieved by incorporating the current contrastive learning framework into LLMs?

To address these questions, we conduct a systematic study by evaluating LLaMA Touvron et al. (2023a) and OPT Zhang et al. (2022) on both semantic textual similarity (STS) tasks and transfer

tasks. Following Jiang et al. (2022), we utilize a prompt such as *This sentence: "* `[text]` *" means* to enable LLMs to generate sentence embeddings, where `[text]` serves as the input slot. This method outperforms traditional representation methods, such as averaging output tokens to represent sentences. Considering the causal architecture and pretraining tasks of LLMs compared to BERT, we can refine the prompt to generate better representations by instructing LLMs to encapsulate as much semantic information of the sentences as possible within the target token.

Inspired by Tsukagoshi et al. (2021), which uses definition sentences from a word dictionary to learn sentence embeddings, we find that performance can be further improved by adding definition sentences and corresponding words as examples to perform in-context learning. To mitigate the gap between examples and input sentences, we also use sentences from the STS-B Cer et al. (2017) training set as examples by instructing ChatGPT to generate a single word to represent the meaning of sentences. By evaluating the demonstration examples based on the STS-B development set, LLMs can outperform previous contrastive learning-based sentence models, which were fine-tuned on unsupervised data.

By scaling up the parameters of LLMs, we find that transitioning from millions to billions of parameters results in improvements on STS tasks. However, continue scaling up may not yield further improvements. Even with in-context learning, 66B OPT still underperforms 6.7B OPT on STS tasks. Nonetheless, scaling up improves performance on transfer tasks. LLMs with tens of billions parameters exhibit strong performances, achieving state-of-the-art performance even without any fine-tuning.

With the advancement of parameter-efficient fine-tuning techniquesHu et al. (2021); Dettmers et al. (2023) and post-training quantization methodsFrantar et al. (2022), we can also fine-tune LLMs with large batch sizes to conduct contrastive learning, even with limited computational resources. For instance, fine-tuning 7B parameter LLMs can be accomplished using the same hardware employed for previous BERT-based models like SimCSE Gao et al. (2021). Even without fine-tuning the full parameters and using the 4-bit quantized method Dettmers et al. (2023), 2.7B OPT with our sentence embeddings method outperforms a 4.8B ST5 Ni et al. (2021) and achieves the state-of-the-art results on STS tasks.

Our main contributions are as follows:

1. We propose a sentence embeddings method that leverages LLMs to enhance the representation of sentences. Additionally, we incorporate in-context learning to further improve the quality of sentence embeddings. Our approach demonstrates that LLMs can generate high-quality sentence embeddings without the need for fine-tuning.

2. We conduct an analysis of scaling up the parameters of LLMs from millions to tens of billions in sentence embeddings. We observe scaling to more than tens of billion parameters may harm the performance on STS tasks. However, the largest model can outperform other counterparts on transfer tasks.

3. Based on our method, we discover that performance can be further enhanced by employing contrastive learning. By adopting efficient fine-tuning techniques, LLMs achieve state-of-the-art performance on STS tasks, even with limited computational resources.

## 2 RELATED WORK

**Sentence Embeddings** Sentence embeddings is to convert a sentence into a fixed-size vector, which captures the semantic meaning and context of the sentence. It allows for the efficient retrieval of similar sentences through the similarity between vectors. Recently, SimCSE Gao et al. (2021) demonstrated that contrastive learning is an effective approach for learning sentence embeddings using BERT in both unsupervised and supervised settings. In the unsupervised setting, SimCSE predicts the input sentence itself from in-batch negatives, with different dropout Srivastava et al. (2014) masks applied. In the supervised setting, Natural Language Inference (NLI) datasets Conneau et al. (2017); Reimers & Gurevych (2019) are used to provide positive and negative pairs. Following the success of SimCSE, there has been a surge of work exploring contrastive learning-based methods. DiffCSE Chuang et al. (2022) incorporates a replaced token detection loss into the contrastive learning framework. PromptBERT Jiang et al. (2022) reveals that prompts can enhance BERT's

ability to represent sentences. Additionally, several studies Cheng et al. (2023); Zhang et al. (2023) have investigated data augmentation for sentence embeddings using LLMs. SentenceT5 (ST5) Ni et al. (2021) leverages the encoder-decoder structure of models, such as T5 Raffel et al. (2020), for generating sentence embeddings and demonstrates improvements by scaling T5 from millions to billions of parameters. However, directly using large language models (LLMs) to generate sentence embeddings remains an area of ongoing research.

**Large Language Models**    LLMs Zhang et al. (2022); Scao et al. (2022); Chowdhery et al. (2022); Touvron et al. (2023a) recently show impressive performance on various natural language process, benefiting from their large parameter sizes compared to previous pretrained language models. LLMs can efficiently learn a new task with in-context learning by using training data as demonstrations Brown et al. (2020). Without any gradient updates, LLMs with in-context learning can solve challenging tasks like multitask language understanding Hendrycks et al. (2020), commonsense reasoning Lin et al. (2021), and math problems Cobbe et al. (2021). This performance can be further improved by scaling up language models Hoffmann et al. (2022); Kaplan et al. (2020).

# 3    METHODOLOGY

In this section, we first discuss current sentence embeddings methods with LLMs, and then introduce a new Prompt-based method with Explicit One word Limitation (PromptEOL) for LLMs in Section 3.1. Based on this method, we describe two settings: without and with fine-tuning. For the setting without fine-tuning, we utilize the in-context learning ability of LLMs to enhance sentence embeddings. To address the issue of lacking textual outputs, we propose two methods to automatically generate demonstrations for in-context learning in Section 3.2. For the setting with fine-tuning, we employ contrastive learning framework, and combine it with the efficient fine-tuning method to alleviate substantial memory requirement in Section 3.3.

## 3.1    REPRESENT SENTENCE WITH LLMS

Previous works Li et al. (2020); Su et al. (2021); Jiang et al. (2022) have extensively studied on improving sentence embeddings from encoder-based pretrained models, like BERT without fine-tuning. Recently, PromptBERT Jiang et al. (2022) leverages a prompt-based method to represent sentence. It uses manual templates like *This sentence: " `[text]` " means* `[MASK]`*.*, where `[text]` is the placeholder for a sentence. The output vector of `[MASK]` token is used as sentence embeddings. It demonstrates superior results compared to previous sentence representation methods like averaging output hidden vectors or the output vector of `[CLS]` token.

Considering to LLMs as autoregression models, which do not have special tokens like `[CLS]` or `[MASK]`, we modify the prompt-based method in Jiang et al. (2022) to make it compatible with LLMs. We use *This sentence: " `[text]` " means* to prompt LLMs generate next token and extract the hidden vectors of the final token as sentence embeddings. To validate the prompt-based method with LLMs, we compare it with two other methods, such as averaging or using the last token as sentence embeddings. For LLMs, we use OPT Zhang et al. (2022) from 125 million parameters to 66 billions and evaluate it on STS-B development set in Figure 1. Following the results in Jiang et al. (2022), we observe that prompt-based method can enhance sentence representation across all OPTs, ranging from millions to billions parameters. Despite that the previous prompt-based method also improved LLMs like OPT on sentence representations, OPT still fails to outperform BERT.

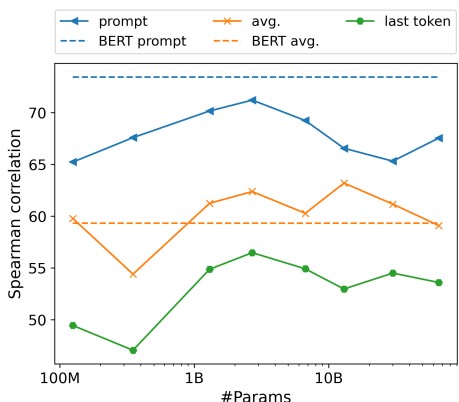

Figure 1: Performances of OPT in STS-B development set with three representation methods. Dash lines represent the results of BERT.

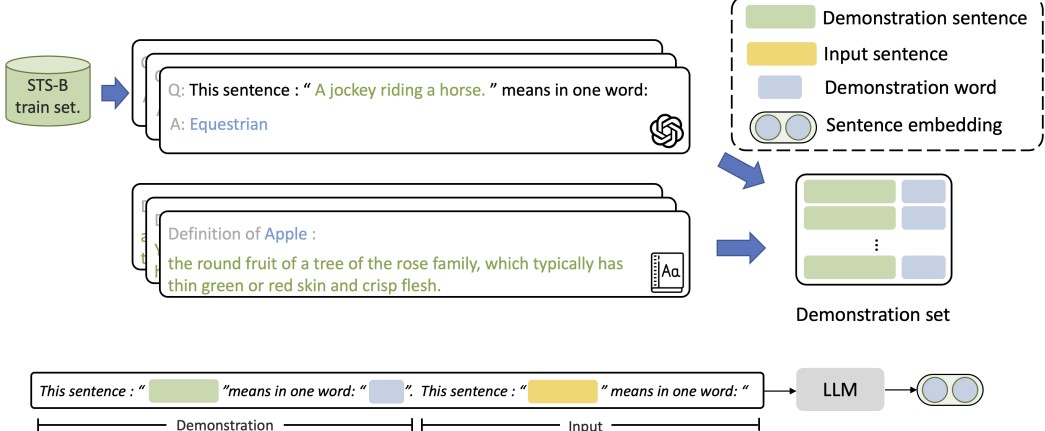

Figure 2: An illustration of in-context learning based sentence embeddings. The green sentences denote the demonstration sentence, and the blue words denote the demonstration words. The corresponding color blocks refer to their slots in the template.

Considering to bidirectional attention in BERT, we hypothesize that BERT can implicitly condense the entire semantic information corresponding to a sentence into a single [MASK] token when using templates like "*This sentence:* " [text] " *means* [MASK].". Since the [MASK] token follows a period, this implicitly restricts BERT to explain meaning into one word. However, this template fails to add the similar "one word limitation" when it is used in autoregression models like OPT with unidirectional attention. To validate this, we simply remove the period in template to transfer it into "*This sentence:* " [text] " *means* [MASK]". Despite only one word difference, and no modification to meaning of the template, the performance of BERT on STS-B development set plummeted from 73.44 to 33.89 Spearman correlation, which means BERT without this implicit "one word limitation" fails to represent sentence.

Inspired by this, our objective is to enhance prompt-based method for LLMs by introducing a "one word limitation". We propose a new Prompt-based method with Explicit One word Limitation (PromptEOL) for LLMs. PromptEOL is simple and straightforward by directly adding some tokens in the template to instruct LLMs in predicting the meaning of sentence in one word. The template we used after modification is following:

*This sentence:* " [text] " *means in one word:* "

Compared to the template in Jiang et al. (2022), we introduce two simple modifications for LLMs. First, we append *in one word* to the prompt to constrain LLMs in predicting semantic information in next token. Secondly, we incorporate *: "* at the end of template to prevent model form generating punctuations in next token, as *This sentence:* " is used to indicate the input of a sentence. We find this template improve all OPT models and allow them to match or even outperform BERT with prompt-based method in Figure 4.

## 3.2 IMPROVE SENTENCE EMBEDDINGS WITH IN-CONTEXT LEARNING

In-context learning is widely utilized as an effective method to help LLMs understand problems. It improves their comprehension of inputs and outputs by directly adding a few examples in the prompts. However, when considering the problem of sentence embeddings, we need to project sentences into vectors based on their semantic information, separately. In other word, sentence embeddings lack textual outputs that could be used as examples to perform in-context learning, such as answers for QA problems or labels for text classification problems. Moreover, there are also no predetermined gold vectors for a given sentence.

To leverage in-context learning in sentence embeddings, we propose an framework to automatically build demonstration sets and search demonstration to improve LLMs sentence embeddings in Figure 2. For the demonstration set, the goal is to create sentence and word pairs, where the word can represents the semantic information of the sentence. We propose two methods to generate pairs.

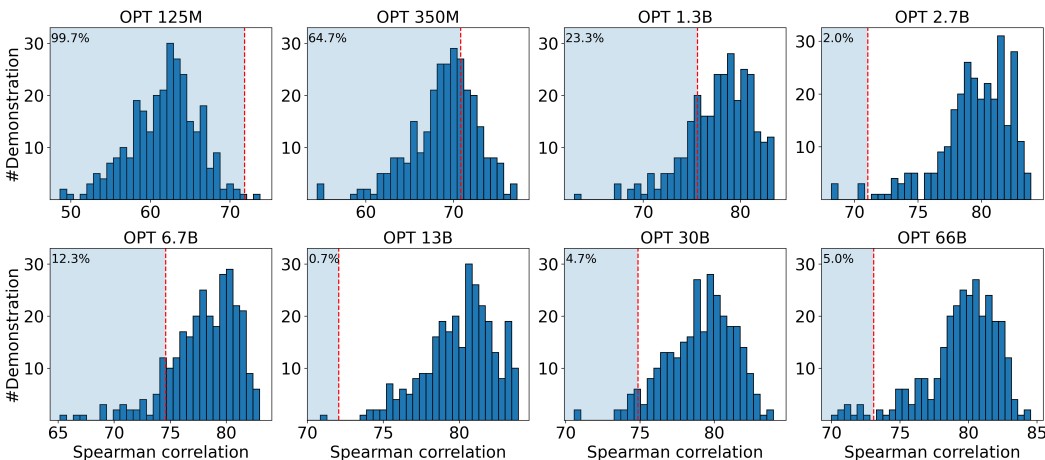

Figure 3: Distribution of Spearman correlations on the STS-B development set with different in-context learning demonstrations. The red dash line represents the Spearman correlation of the corresponding model without any demonstration. The blue area represents demonstrations that negatively impact the performance, and the percentage refers to the proportion of these demonstrations to the total number of demonstrations.

The first method involves using ChatGPT to generate corresponding words according to the semantic information of given sentences from STS-B training set. By asking ChatGPT with same template in Figure 2, ChatGPT outputs one word summary for the given sentence. We also find "one word limitation" in Section 3.1 is important for ChatGPT. Consider to our prompt-based representation method, we employ the hidden state of the next token as the sentence embeddings. By removing *in one word* from the template, it tends to explain the meaning of a sentence in a lengthy way, and the first word often becomes an article such as "The", which lacks clear meaning. For example, given the sentence "A jockey riding a horse.", the hidden state achieves the highest dot product similarity for "Equestrain" among its word embeddings. However, without "one word limitation", it will achieve the highest dot product similarity for word without specific meaning such as "The" among its word embeddings, which can not represent sentence properly. Inspired by DefSent Tsukagoshi et al. (2021), which leverages definition sentences with their words as labels to train unsupervised sentence embedding, our second method is also based on a word dictionary. We directly use words and their definition sentences in the Oxford dictionary as word-sentence pairs.

Based on these methods, we construct a demonstration set consisting of 300 pairs of sentences and words. 100 pairs are from STS-B training set, with words labeled by ChatGPT, while the remaining are from the Oxford dictionary. To find demonstration that help model to represent sentences, we directly evaluate each demonstration on the STS-B development set and use the demonstration with the best Spearman correlation as the demonstration for corresponding models. We also visualize the distribution of Spearman correlations for OPT from 125M to 66B parameters in Figure 3. Following the previous study Kaplan et al. (2020), we notice that in-context learning achieves better performance, when increasing model parameter from 125M to 2.7B. For example, there are only one demonstration that helps the 125M OPT achieve better performance compared to without demonstration. However, around 98% of demonstrations improve the performance of the 2.7B OPT. In-context learning significantly enhance the sentence embeddings, especially for OPT with more than 1B parameters. With only in-context learning, OPT with more than 1.3B parameters even achieve better results on STS tasks compared to contrastive learning based method like SimCSE Gao et al. (2021) in Table 1.

## 3.3 Contrastive Learning with Efficient Fine-tuning

Since in-context learning boosts sentence embeddings performances without any gradient update, we also exploit contrastive learning on LLMs, which has been demonstrated as an efficient way to learn sentence embeddings Gao et al. (2021). It can be divided into unsupervised and supervised settings, according to the datasets. For unsupervised setting, the sentences in dataset lack corresponding positive and negative sentences to perform contrastive learning. For supervised setting, natural language inference (NLI) datasets are used as the datasets, and each sentence has corresponding

positive and negative sentences. In this section, we focus on the supervised setting to fully leverage LLMs for sentence embeddings.

However, contrastive learning requires a large batch size to increase the number of negative samples, which demands a high amount of GPU memory, especially in the supervised setting. For example, SimCSE uses a batch size of 512 to fine-tune 110M BERT in the supervised setting. Each batch includes 1536 sentences, containing both their positive and hard negative sentences. It requires 58GB of GPU memory on 4 GPUs. As a result, fine-tuning LLMs with contrastive learning becomes challenging due to the memory requirements, particularly for models with significantly larger parameter sizes than BERT.

To solve this problem, we leverage current efficient fine-tuning method QLoRA Dettmers et al. (2023). QLoRA combines two techniques to significantly reduces memory usage: 4-bit quantization and parameter efficient fine-tuning. Quantization reduces the memory usage of LLMs by quantizing their weight from 16-bit to 4-bit. Parameter efficient fine-tuning with LoRA Hu et al. (2021) significantly reduces the memory usage of optimizer compared to full fine-tuning by only fine-tuning small proportion of weight.

Following Gao et al. (2021), we use SNLI and MNLI datasets where each sentence $x_i$ has corresponding a positive sentence $x_i^+$ and a hard negative sentence $x_i^-$. To represent sentence, we use our prompt-based method in Section 3.1. Formally, given sentence $x_i$, we first add $x_i$ to the template and get hidden states:

$$\mathbf{h}_{i1}, \ldots, \mathbf{h}_{il} = \text{LLM}(\textit{This sentence: ``}x_i\textit{'' means in one word: ``}) \tag{1}$$

where $l$ is the number of hidden states. We then use last token hidden state as its sentence embedding $\mathbf{h}_i = \mathbf{h}_{il}$. Since we can represent the sentence pair $(x_i, x_i^+, x_i^-)$ to their embeddings $(\mathbf{h}_i, \mathbf{h}_i^+, \mathbf{h}_i^-)$. Our training objective is following:

$$\ell_i = -\log \frac{e^{\cos(\mathbf{h}_i, \mathbf{h}_i^+)/\tau}}{\sum_{j=1}^{N} \left( e^{\cos(\mathbf{h}_i, \mathbf{h}_j^+)/\tau} + e^{\cos(\mathbf{h}_i, \mathbf{h}_j^-)/\tau} \right)} \tag{2}$$

where $N$ is the batch size and $\tau$ is the temperature hyperparameter in contrastive learning.

## 4 EXPERIMENT

### 4.1 IMPLEMENTATION DETAILS

For the setting without fine-tuning, we use OPT from 125M to 66B parameters, and LLaMA from 7B to 65B parameters. All models use the same template in Section 3.1. We use 300 pairs of sentences and words as demonstration set for in-context learning. Among these, 100 pairs are from the STS-B training set, and we use `gpt-3.5-turbo` to label their words. The remaining 200 pairs are from the Oxford dictionary. We provide all demonstrations in Appendix B. For each model, we choose only one demonstration that has the highest Spearman correlation on the STS-B development set as their demonstration for evaluation. All results from models with 16-bit weights. We also present results using quantization methods in Appendix C.

For the setting with fine-tuning, we use QLoRA Dettmers et al. (2023) to fine-tune OPT and LLaMA with contrastive learning. Following QLoRA, we use LoRA $r = 64, \alpha = 16$, dropout $= 0.05$, and add LoRA modules on all linear layers of the 4-bit quantized model. We fine-tune models on the NLI datasets Gao et al. (2021) with one epoch, temperature $\tau = 0.05$ and learning rate 5e-4. Due to hardware limitations, we only conduct our experiments with model parameters less than or equal to 13B with 8 RTX-3090 GPUs. For models with fewer than 7B parameters, we fine-tune them on 2 GPUs with a batch size of 256. For 7B models, we use 4 GPUs with a batch size of 256. For 13B models, we use 8 GPUs with a batch size of 200.

### 4.2 DATASET

Following previous works Gao et al. (2021); Jiang et al. (2022), We use the SentEval toolkit Conneau & Kiela (2018) to conduct our experiments on seven STS datasets and seven transfer learning datasets.

| Method | Params | STS12 | STS13 | STS14 | STS15 | STS16 | STS-B | SICK-R | Avg. |
|---|---|---|---|---|---|---|---|---|---|
| | | | | *Fine-tuning on unsupervised datasets* | | | | | |
| SimCSE-BERT[†] | 110M | 68.40 | 82.41 | 74.38 | 80.91 | 78.56 | 76.85 | 72.23 | 76.25 |
| SimCSE-RoBERTa[†] | 123M | 70.16 | 81.77 | 73.24 | 81.36 | 80.65 | 80.22 | 68.56 | 76.57 |
| PromptBERT[‡] | 110M | 71.56 | 84.58 | 76.98 | 84.47 | 80.60 | 81.60 | 69.87 | 78.54 |
| PromptRoBERTa[‡] | 123M | 73.94 | 84.74 | 77.28 | 84.99 | 81.74 | 81.88 | 69.50 | 79.15 |
| | | | | *Without fine-tuning* | | | | | |
| BERT avg.[†] | 110M | 30.87 | 59.89 | 47.73 | 60.29 | 63.73 | 47.29 | 58.22 | 52.57 |
| BERT prompt[‡] | 110M | 60.96 | 73.83 | 62.18 | 71.54 | 68.68 | 70.60 | 67.16 | 67.85 |
| ST5-Enc[§] | 4.8B | 34.97 | 60.19 | 47.59 | 66.40 | 70.62 | 62.83 | 63.57 | 58.02 |
| | 125M | 59.90 | 71.55 | 60.93 | 70.76 | 72.83 | 67.89 | 65.14 | 67.00 |
| | 350M | 54.70 | 71.52 | 59.99 | 64.51 | 71.39 | 66.55 | 66.58 | 65.03 |
| | 1.3B | 64.59 | 79.06 | 68.46 | 78.88 | 78.64 | 73.22 | 69.41 | 73.18 |
| PromptEOL | 2.7B | 60.03 | 75.51 | 64.30 | 74.56 | 77.62 | 67.73 | 65.35 | 69.30 |
| OPT | 6.7B | 60.91 | 80.05 | 67.65 | 75.49 | 80.11 | 72.91 | 67.57 | 72.10 |
| | 13B | 60.21 | 81.36 | 69.69 | 75.46 | 79.58 | 70.73 | 65.99 | 71.86 |
| | 30B | 59.99 | 80.52 | 69.80 | 75.20 | 78.03 | 73.57 | 69.87 | 72.43 |
| | 66B | 55.66 | 74.62 | 64.90 | 72.34 | 75.21 | 71.72 | 67.43 | 68.84 |
| | 125M | 62.22 | 73.10 | 61.84 | 71.09 | 72.08 | 67.80 | 64.10 | 67.46 |
| | 350M | 63.87 | 73.85 | 63.41 | 72.45 | 73.13 | 70.84 | 65.61 | 69.02 |
| | 1.3B | 72.78 | 83.77 | 73.61 | 83.42 | 80.60 | 78.80 | 69.69 | 77.52 |
| PromptEOL+ICL | 2.7B | 68.49 | 84.72 | 75.15 | 83.62 | 81.34 | 80.94 | 72.97 | 78.18 |
| OPT | 6.7B | 70.65 | 84.51 | 75.01 | 83.51 | 82.00 | 81.12 | 76.77 | 79.08 |
| | 13B | 71.99 | 85.22 | 76.04 | 82.23 | 81.38 | 81.42 | 75.00 | 79.04 |
| | 30B | 69.99 | 83.35 | 74.75 | 83.14 | 82.42 | 81.45 | 77.46 | 78.94 |
| | 66B | 69.93 | 83.29 | 74.88 | 80.10 | 81.11 | 81.76 | 76.26 | 78.19 |

Table 1: Performances of our method on STS tasks without fine-tuning. ICL denotes in-context learning with our demonstration set. †: results from Gao et al. (2021). ‡: results from Jiang et al. (2022). §: results from Ni et al. (2021). More results on other LLMs can be found in Appendix F.

The STS datasets include STS tasks 2012-2016 Agirre et al. (2012; 2013; 2014; 2015; 2016) STS-B Cer et al. (2017), SICK-R Marelli et al. (2014). Sentence pairs in each STS dataset are scored from 0 to 5 to indicate semantic similarity. Spearman correlation is used as a metric to evaluate the correlation between the cosine similarity of sentence embeddings and the golden similarity scores. The transfer learning datasets include MR Pang & Lee (2005), CR Hu & Liu (2004), SUBJ Pang & Lee (2004), MPQA Wiebe et al. (2005), SST-2 Socher et al. (2013), TREC Voorhees & Tice (2000) and MRPC Dolan & Brockett (2005). Sentence embeddings are used as input feature to train corresponding logistic regression classification.

## 4.3 RESULTS

We compare our method with BERT-based methods such as SBERT Reimers & Gurevych (2019), SimCSE Gao et al. (2021), and PromptBERT Jiang et al. (2022). In addition, we include other sentence methods based on LLMs as baselines, such as ST5 Ni et al. (2021) and SGPT Muennighoff (2022). Among these baselines, ST5 achieves state-of-the-art results on both STS and transfer learning tasks by further fine-tuning 4.8B parameters T5 encoder with contrastive learning.

**STS tasks without fine-tuning** Table 1 shows the results of PromptEOL with and without in-context learning on STS tasks. Even without corresponding textual outputs for sentence embeddings, in-context learning still helps model to generate better embeddings. As the model size grows, improvements from in-context learning also increase. Moreover, in-context learning shows significantly improvements on STS tasks for model with more than billions parameters. For instances, it raises the Spearman correlation from 68.84 to 78.19 on 66B OPT. Our method with in-context learning also outperforms among methods without fine-tuning. Even if we do not use any method to avoid anisotropy Ethayarajh (2019), which is widely regarded as the main reason for poor performance on STS tasks Gao et al. (2021); Ni et al. (2021), our method still outperforms unsupervised methods such as SimCSE and PromptBERT, which use contrastive learning to avoid anisotropy. Additionally,

| Method | Params | STS12 | STS13 | STS14 | STS15 | STS16 | STS-B | SICK-R | Avg. |
|---|---|---|---|---|---|---|---|---|---|
| *Fine-tuning on supervised datasets* | | | | | | | | | |
| SBERT-NLI[†] | 220M | 72.27 | 78.46 | 74.90 | 80.99 | 76.25 | 79.23 | 73.75 | 76.55 |
| SimCSE-RoBERTa[†] | 123M | 76.53 | 85.21 | 80.95 | 86.03 | 82.57 | 85.83 | 80.50 | 82.52 |
| | 354M | 77.46 | 87.27 | 82.36 | 86.66 | 83.93 | 86.70 | 81.95 | 83.76 |
| PromptRoBERTa[‡] | 123M | 76.75 | 85.93 | 82.28 | 86.69 | 82.80 | 86.14 | 80.04 | 82.95 |
| SGPT[¶] | 5.8B | 74.28 | 85.35 | 79.21 | 85.52 | 82.54 | 85.50 | 79.53 | 81.70 |
| ST5-Enc[§] | 4.8B | 80.10 | 88.75 | 84.70 | 88.86 | 85.17 | 86.77 | 80.39 | 84.96 |
| PromptEOL+CSE OPT | 1.3B | 79.01 | 89.26 | 84.10 | 88.30 | 84.62 | 87.71 | 80.52 | 84.79 |
| | 2.7B | 79.49 | 89.64 | 84.80 | 89.51 | 85.91 | 88.33 | 81.64 | 85.62 |
| | 6.7B | 80.14 | 90.02 | 84.94 | 89.78 | 85.84 | 88.75 | 81.29 | 85.82 |
| | 13B | 80.20 | 90.24 | 85.34 | 89.52 | 85.90 | 88.56 | 82.06 | 85.97 |
| PromptEOL+CSE LLaMA | 7B | 79.16 | 90.22 | 85.40 | 88.99 | 86.25 | 88.37 | 81.51 | 85.70 |
| | 13B | 78.63 | 90.03 | 85.46 | 89.48 | 86.18 | 88.45 | 82.69 | 85.85 |

Table 2: Performances of our method on STS tasks with fine-tuning. CSE denotes contrastive learning for sentence embeddings. †: results from Gao et al. (2021). §: results from Ni et al. (2021). ¶: results from evaluation the public checkpoint Muennighoff (2022) on STS tasks.

| Method | Params | MR | CR | SUBJ | MPQA | SST | TREC | MRPC | Avg. |
|---|---|---|---|---|---|---|---|---|---|
| *Fine-tuning on supervised datasets* | | | | | | | | | |
| SimCSE-RoBERTa[†] | 123M | 84.92 | 92.00 | 94.11 | 89.82 | 91.27 | 88.80 | 75.65 | 88.08 |
| | 220M | 81.12 | 92.37 | 95.11 | 90.49 | 92.75 | 91.80 | 76.64 | 89.61 |
| PromptRoBERTa[‡] | 123M | 85.74 | 91.47 | 94.81 | 90.93 | 92.53 | 90.40 | 77.10 | 89.00 |
| ST5-Enc[§] | 4.8B | 90.83 | 94.44 | 96.33 | 91.68 | 94.84 | 95.40 | 77.91 | 91.63 |
| *Without fine-tuning* | | | | | | | | | |
| BERT avg. | 110M | 78.66 | 86.25 | 94.37 | 88.66 | 84.40 | 92.80 | 69.54 | 84.94 |
| ST5-Enc[§] | 4.8B | 91.15 | 93.33 | 97.55 | 90.20 | 94.07 | 94.40 | 74.26 | 90.71 |
| PromptEOL OPT | 1.3B | 88.06 | 91.55 | 95.90 | 91.55 | 93.08 | 95.00 | 73.97 | 89.87 |
| | 2.7B | 88.83 | 92.29 | 95.93 | 91.76 | 94.62 | 96.00 | 75.94 | 90.77 |
| | 6.7B | 90.26 | 92.50 | 96.67 | 91.39 | 94.67 | 96.00 | 77.91 | 91.34 |
| | 13B | 90.73 | 92.90 | 96.69 | 91.48 | 94.01 | 96.80 | 75.59 | 91.17 |
| | 30B | 90.95 | 92.77 | 96.99 | 91.79 | 95.28 | 97.00 | 73.97 | 91.25 |
| | 66B | 90.96 | 93.40 | 97.01 | 91.93 | 95.22 | 96.40 | 75.25 | 91.45 |
| PromptEOL LLaMA | 7B | 90.40 | 92.90 | 96.88 | 91.57 | 95.11 | 95.40 | 75.13 | 91.06 |
| | 13B | 92.02 | 93.22 | 97.29 | 91.40 | 95.66 | 95.80 | 76.46 | 91.69 |
| | 30B | 91.64 | 93.27 | 97.10 | 91.86 | 95.99 | 95.80 | 78.43 | 92.01 |
| | 65B | 92.13 | 93.43 | 97.16 | 91.91 | 95.33 | 97.40 | 77.28 | 92.09 |

Table 3: Performances of our method on transfer learning tasks. †: results from Gao et al. (2021). ‡: results from Jiang et al. (2022). §: results from Ni et al. (2021).

we find the performance is not sensitive to the model size while scaling model beyond a billion parameters. Smaller models, such as 1.3B OPT, even outperforms SimCSE without fine-tuning.

**STS tasks with fine-tuning** Table 2 shows the results by fine-tuning with PromptEOL on the supervised dataset. Compared to ST5-Enc, which fine-tuned all 4.8B parameters on Community QA and NLI datasets, our method with 2.7B OPT achieves superior results through parameter-efficient fine tuning on the 4-bit model with only NLI datasets. Keep scaling up the parameters size, 13B OPT and LLaMA achieve the best performance on STS tasks. However, the improvement in scaling model parameters from 2.7B to 13B is not significant.

**Transfer tasks** We also report the results of our method on the transfer learning tasks in Table 3. Unlike STS tasks, we observe that LLMs achieve better performance as the model size increases. Specifically, the 66B OPT and 65B LLaMA models outperform their smaller counterparts with our representation method. Based on our representation method, LLMs show good performance without in-context learning and contrastive learning. Following ST5 Ni et al. (2021), we find that applying

contrastive learning solely on NLI datasets can even harm performance on transfer tasks. To solve this problem, ST5 utilizes additional datasets, such as the Community QA dataset, to enhance its performance in transfer tasks. For in-context learning, as it is widely used in text classification, we find that using examples not relevant to tasks, such as STS-B or the dictionary, does not enhance transfer task performance. We present these results in Appendix D.

# 5 ANALYSIS

## 5.1 SENTENCE REPRESENTATION METHODS

We present the results obtained using three sentence representation methods, across models ranging in size from 125M to 66B parameters, as shown in Figure 4. Different representation methods can yield significantly different results. Prompt-based methods outperform direct averaging in three settings. Among these methods, PromptEOL exhibits the best performance, as it introduces an explicit "one-word limitation". More detail results can be found in Appendix E.

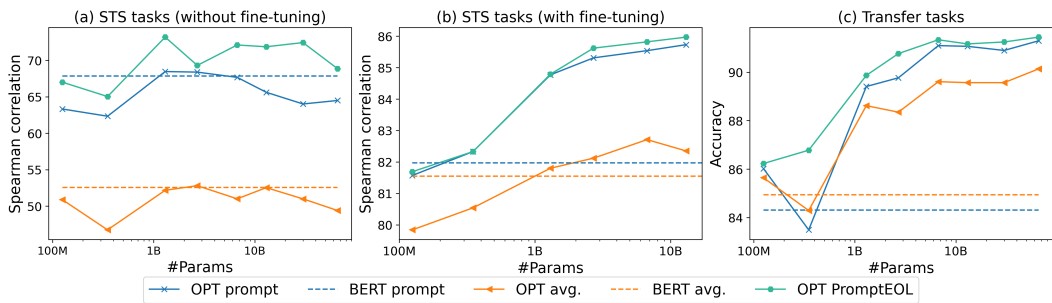

Figure 4: Influence of different sentence representation methods on three settings. "avg." refers to use averaging output tokens as sentence embeddings. "prompt" refers to extract sentence embeddings using the template from Jiang et al. (2022) . Dash lines represent the results from the base-size BERT.

## 5.2 IN-CONTEXT LEARNING

We demonstrate in-context learning examples that were obtained from each model on the STS-B development set, along with corresponding improvements on Spearman correlation for STS tasks. As the size of the model increases to 2.7B, the improvements in in-context learning become more and more pronounced, and related examples are usually more implicit. For instance, the 125M OPT uses examples where words are incorporated within the sentence.

|       | Sentence | Word | Improve |
|-------|----------|------|---------|
| 125M  | A man is smoking. | Smoking | 0.46 |
| 350M  | A man is playing on a guitar and singing. | Music | 3.99 |
| 1.3B  | relating to switzerland or its people. | Swiss | 4.34 |
| 2.7B  | A jockey riding a horse. | Equestrian | 8.88 |
| 6.7B  | The man is riding a horse. | Horseback-riding | 6.98 |
| 13B   | meat from a deer. | Venison | 7.18 |
| 30B   | The man is riding a motorcycle down the road. | Motorcycling | 6.51 |
| 66B   | of or relating to tutors or tutoring. | Tutorial | 9.35 |

Table 4: In-context learning examples used in various model size.

# 6 CONCLUSION

In this paper, we focus on exploiting Large Language Models (LLMs) to improve sentence embeddings. To achieve this, we propose a new sentence embeddings method called PromptEOL, which adapts previous prompt-based methods to autoregression models. Furthermore, we leverage in-context learning to generate superior sentence embeddings by utilizing ChatGPT and the Oxford dictionary to create sentence embeddings demonstrations. It demonstrates in-context learning allows LLMs to achieve performance comparable to current contrastive learning methods. With our promtp-based method, we also discover that further fine-tuning of LLMs can achieve the state-of-the-art performance using only efficient fine-tuning methods.

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

## A  APPENDIX

You may include other additional sections here.

## B  DEMONSTRATIONS

| | |
|---|---|
| Over 100 dead as typhoon slams central Philippines. | Disaster |
| Woman in red overalls standing on the sidewalk. | Observation |
| India starts voting in world's largest election. | Democracy |
| Three dogs pulling a man on a bicycle through the snow. | Adventure |
| Spain approves new restrictive abortion law. | Legislation |
| A man dives into a pool. | Activity |
| Saudi to give Lebanese army $3 billion | Aid |
| Updated - Two explosions near finish line of Boston Marathon | Terrorism |
| A gray cat with green eyes looks at the camera. | Portrayal |
| Egypt interior minister survives bomb | Survival |
| A man is playing a large flute. | Music |
| A man is spreading shreded cheese on a pizza. | Cooking |
| Three men are playing chess. | Strategy |
| A man is playing the cello. | Music |
| Some men are fighting. | Conflict |
| A man is smoking. | Smoking |
| The man is playing the piano. | Music |
| A man is playing on a guitar and singing. | Music |
| A person is throwing a cat on to the ceiling. | Cruelty |
| The man hit the other man with a stick. | Violence |
| A woman picks up and holds a baby kangaroo. | Caring |
| A man is playing a flute. | Music |
| A person is folding a piece of paper. | Origami |
| A man is running on the road. | Exercise |
| A dog is trying to get bacon off his back. | Humorous |
| The polar bear is sliding on the snow. | Playful |
| A woman is writing. | Writing |
| A cat is rubbing against baby's face. | Affection |
| The man is riding a horse. | Horseback-riding |
| A man pours oil into a pot. | Cooking |
| A man is playing a guitar. | Music |
| A panda is sliding down a slide. | Playful |
| A woman is eating something. | Eating |
| A woman peels a potato. | Cooking |
| The boy fell off his bike. | Accident |
| The woman is playing the flute. | Music |
| A rabbit is running from an eagle. | Escape |
| The woman is frying a breaded pork chop. | Cooking |

| | |
|---|---|
| A girl is flying a kite. | Recreation |
| A man is riding a mechanical bull. | Entertainment |
| The man is playing the guitar. | Music |
| A woman is dancing and singing with other women. | Celebration |
| A man is slicing a bun. | Cooking |
| A man is pouring oil into a pan. | Cooking |
| A lion is playing with people. | Dangerous |
| A dog rides a skateboard. | Unusual |
| Someone is carving a statue. | Art |
| A woman is slicing an onion. | Cooking |
| A woman is dancing. | Dancing |
| Two green and white trains sitting on the tracks. | Arrangement |
| A small white cat with glowing eyes standing underneath a chair. | Mysterious |
| A large boat in the water at the marina. | Yacht |
| a bus driving in a street. | Movement |
| A passenger train waiting in a station. | Stationary |
| a woman at a dinner table writing on her notebook. | Observation |
| An Apple computer sitting on the floor. | Description |
| A close-up of a brown horse's head. | Detail |
| A group of people eat at a table outside. | Alfresco |
| A jockey riding a horse. | Equestrian |
| The man is riding a motorcycle down the road. | Motorcycling |
| A woman riding a brown horse. | Equestrian |
| A kid jumping a ledge with a bike. | Stunt |
| A black dog standing in front of yellow flowers. | Contrast |
| Close up of a bottle of water. | Zoom |
| A close up of a brown faced cat. | Intense |
| sheep standing in afield. | Pastoral |
| A longed-haired cat with it's eyes closed. | Sleeping |
| A woman in a gray shirt smiles for the camera while the woman behind her makes a face. | Contrast |
| A silver and blue Amtrak train on the tracks near a small train station. | Railway |
| A person in a blue shirt reclines near a coffee table and television. | Relaxation |
| A black and white photo of a woman showing a horse. | Monochrome |
| A dark brown horse standing in a field. | Equine |
| A pitched tent with a horse in the background. | Camping |
| A group of people sitting around a table with food on it. | Gathering |
| A brown horse stands in a lush green field. | Pastoral |
| a black and white cow in hay. | Cow |
| An elderly woman stands in a kitchen with two cats at her feet. | Domesticity |
| A school bus is driving uphill on a rural road. | Ascend |
| Camouflage airplane sitting on grassy field. | Concealment |
| Three young women standing in a room together. | Group |
| Red double decker bus driving through the streets. | Transportation |
| A white sheep on a hillside looking at the camera. | Observation |
| A group of sheep in a field. | Flock |
| A close-up, distorted photo of an empty glass Coke bottle. | Abstract |
| Very crowded office desk with computer monitor on. | Cluttered |
| A man sitting in a cluttered room. | Disorderly |
| Two white cows in a green pasture. | Scene |
| Black cow walking under trees in pasture. | Nature |
| Two people sitting at a table at a restaurant. | Dining |
| A smiling woman with a beer sitting outside with another smiling woman. | Companionship |
| A bird holding on to a metal gate. | Perching |
| The skinny cows are standing on the grass. | Cattle |
| A women laying across two men sitting on a sofa. | Entanglement |
| a woman with a big necklace. | Opulent |
| Brown cow with horns standing in a field. | Cattle |
| A cruise liner docked at the shoreline. | Berthed |

| | |
|---|---|
| Black and white cat lying under bush. | Camouflage |
| Brown and white cow standing in grass at side of road. | Cow |
| A small dog looking up at the camera while standing on grass. | Adorable |
| the process or result of becoming smaller or pressed together. | Contraction |
| done, produced, or occurring once a week. | Weekly |
| the chief bishop of an eparchy. | Eparch |
| a native or inhabitant of guatemala, or a person of guatemalan descent. | Guatemalan |
| the energy transmitted by radiation. | Radiation |
| a necktie tied in a loose knot with two hanging ends, popular in the late 19th and early 20th centuries. | Four-in-hand |
| relating to germany, its people, or their language. | German |
| not yet used or soiled. | Fresh |
| the chemical composition and properties of a substance or body. | Chemistry |
| insects of the order Hemiptera; true bugs. | Hemiptera |
| an act of counting something again, especially votes in an election. | Recount |
| a very helpful or valuable event, person, or article. | Godsend |
| the part of a theatre where the orchestra plays, typically in front of the stage and on a lower level. | Orchestra |
| the eighth star in a constellation. | Theta |
| abnormally low blood pressure. | Hypotension |
| high-flown style; excessive use of verbal ornamentation. | Rhetoric |
| impetuous or flamboyant vigour and confidence; panache. | Dash |
| a large and densely populated urban area; may include several independent administrative districts. | Metropolis |
| the side of an object that is opposite its front. | Backside |
| an outward semblance that misrepresents the true nature of something. | Disguise |
| the action of reasserting or confirming something. | Reaffirmation |
| an idea or conclusion having general application. | Generalization |
| the choicest or most essential or most vital part of some idea or experience. | Nub |
| the way in which something is done or operated. | Mechanics |
| relating to switzerland or its people. | Swiss |
| an inhabitant of a particular town or city. | Citizen |
| a compound present in some kinds of ergot. an alkaloid, it causes constriction of blood vessels and is used in the treatment of migraine. | Ergotamine |
| the descendants of one individual. | Parentage |
| things done to express interest in or please someone. | Attention |
| the branch of technology that deals with dimensions and tolerances of less than 100 nanometres, especially the manipulation of individual atoms and molecules. | Nanotechnology |
| a printed heading on stationery, stating a person or organization's name and address. | Letterhead |
| people who are destined to die soon. | Doomed |
| the cross on which christ was crucified. | Cross |
| a member of a sect. | Sectary |
| an inanimate object worshipped for its supposed magical powers or because it is considered to be inhabited by a spirit. | Fetish |
| denoting the offspring of a cross. | Filial |
| create or prepare methodically. | Formulate |
| a small old world songbird of the thrush family, with black, white, and brown coloration and a harsh call. | Chat |
| make oneself thinner by dieting and sometimes exercising. | Slim |
| head into a specified direction. | Make |
| a white new zealander as opposed to a maori. | Pakeha |
| a place of inviolable privacy. | Sanctum |
| a person who has matriculated. | Matriculate |
| agriculture developed along industrial lines. | Agro-industry |
| a naval officer of the second most senior rank, above vice admiral and below admiral of the fleet or fleet admiral. | Admiral |
| ease the grief or distress of. | Comfort |
| come under, be classified or included. | Fall |
| be a sign or indication of. | Denote |

| | |
|---|---|
| the starting point for a new state or experience. | Threshold |
| an instance of sleeping in rough accommodation or on an improvised bed. | Doss |
| a writer of any of the hagiographa. | Hagiographer |
| relating to or denoting a paraprofessional. | Paraprofessional |
| intense and eager enjoyment, interest, or approval. | Enthusiasm |
| kill and prepare for market or consumption. | Dress |
| an unexpected and surprising event, especially an unpleasant one. | Bombshell |
| obtain or seek to obtain by cadging or wheedling. | Scrounge |
| a mechanical device consisting of a cylindrical tube around which the hair is wound to curl it. | Crimper |
| an established ceremony prescribed by a religion. | Rite |
| a continuous period of being seated, especially when engaged in a particular activity. | Sitting |
| the cultivation of flowers. | Floriculture |
| settle or establish firmly. | Cement |
| meat from a deer. | Venison |
| a deep red colour like that of burgundy wine. | Burgundy |
| a temporary board fence erected round a building site. | Hoarding |
| haunt like a ghost; pursue. | Obsess |
| the quality of transparency or purity. | Clarity |
| a push or blow, especially one given with the head. | Butt |
| a standard or typical example. | Paradigm |
| praise enthusiastically and publicly. | Acclaim |
| pass through a hole or opening. | Reeve |
| relating to or characteristic of java, a large island in the malay archipelago. | Javan |
| a substance obtained by mining. | Mineral |
| the solid part of a comet's head. | Nucleus |
| confine or restrain with or as if with manacles or handcuffs. | Manacle |
| cause extensive destruction or ruin utterly. | Devastate |
| a person being dealt with by social or medical services. | Client |
| make or become very warm, especially through exposure to the heat of the sun or a fire. | Roast |
| say something with difficulty, repeating the initial consonants of words. | Stutter |
| a body of students who are taught together. | Class |
| euphemistic expressions for death. | Release |
| of or relating to or resembling fish. | Fishy |
| the part of a sphere cut off by any plane not passing through the centre. | Segment |
| a crossbar in front of a wagon with a swingletree at each end, enabling two horses to be harnessed. | Doubletree |
| a strong blow with a knife or other sharp pointed instrument. | Thrust |
| a shiny silicate mineral with a layered structure, found as minute scales in granite and other rocks, or as crystals. it is used as a thermal or electrical insulator. | Mica |
| coins or other articles made of gold. | Gold |
| living quarters provided for public convenience. | Accommodation |
| unwillingness to do something contrary to your custom. | Loath |
| move or cause to move gradually or with difficulty into another position. | Work |
| move or sway in a rising and falling or wavelike pattern. | Fluctuate |
| a flexible covering for the base of a gear lever or other mechanical part. | Gaiter |
| done or existing alone. | Solitary |
| of or relating to tutors or tutoring. | Tutorial |
| come or be in close contact with; stick or hold together and resist separation. | Cling |
| swell or cause to swell. | Belly |
| relating to mongolia, its people, or their language. | Mongolian |
| a longing or yearning. | Yen |
| the sound made by the vibration of vocal folds modified by the resonance of the vocal tract. | Vocalisation |
| the neurophysiological processes, including memory, by which an organism becomes aware of and interprets external stimuli. | Perception |
| the process or action by which something is reabsorbed. | Resorption |
| a public statement containing information about an event that has happened or is going to happen. | Promulgation |

| | |
|---|---|
| in an advanced stage of pregnancy. | Heavy |
| a smoky outdoor fire that is lit to keep off insects or protect plants against frost. | Smudge |
| direct in spatial dimensions; proceeding without deviation or interruption; straight and short. | Direct |
| a dead body, especially of a human being rather than an animal. | Corpse |
| distinctive and stylish elegance. | Style |
| a very typical example of a certain person or thing. | Archetype |
| a person who replies to something, especially one supplying information for a questionnaire or responding to an advertisement. | Respondent |
| the action of entering something. | Entry |
| on the italian or roman side of the alps. | Ultramontane |
| a projecting piece of wood made for insertion into a mortise in another piece. | Tenon |
| a display of pretended or exaggerated suffering to obtain sympathy. | Martyrdom |
| a malevolent spirit or person. | Cacodemon |
| something or someone that causes anxiety; a source of unhappiness. | Vexation |
| impose or inflict forcefully. | Clamp |
| a long essay on a particular subject, especially one written for a university degree or diploma. | Dissertation |
| be close or similar. | Approximate |
| of uncertain outcome; especially fraught with risk. | Chancy |
| the brotherhood of freemasons. | Craft |
| a supporter of the american side during the war of american independence. | Whig |
| a formal document giving notice of your intention to resign. | Resignation |
| a device used in taxis that automatically records the distance travelled and the fare payable. | Taximeter |
| any long object resembling a thin line. | Thread |
| a set of reasons or a logical basis for a course of action or belief. | Rationale |
| a person appointed to select a representative team in a sport. | Selector |
| the manner in which someone behaves towards or deals with someone or something. | Treatment |
| refuse to acknowledge someone or something as having authority. | Revolt |
| a branch of an army assigned to a particular kind of work. | Corps |
| an event resulting in great loss and misfortune. | Cataclysm |
| occupy or take on. | Strike |
| move with sweeping, effortless, gliding motions. | Sweep |
| a high point, level, or figure. | High |
| a large luxurious passenger ship of a type formerly used on a regular line. | Liner |
| more distant than another object of the same kind. | Far |
| the underground lair of a badger or fox. | Earth |
| the central principle or part of a policy, system, etc., on which all else depends. | Keystone |
| chequer with contrasting colours. | Counterchange |
| the condition of being fenestrate. | Fenestration |
| observe with care or pay close attention to. | Observe |
| a dark greenish-blue colour. | Teal |
| a mystic syllable, considered the most sacred mantra in hinduism and tibetan buddhism. it appears at the beginning and end of most sanskrit recitations, prayers, and texts. | Om |
| set the level or character of. | Gear |
| be sexually unfaithful to one's partner in marriage. | Betray |
| a round button for adjusting or controlling a machine. | Knob |
| an army unit consisting of soldiers who fight on foot. | Foot |
| people who are fearful and cautious. | Timid |
| the trait of being excessively fastidious and easily shocked. | Squeamishness |
| demand something forcefully, not accepting refusal. | Insist |
| a secret word or phrase known only to a restricted group. | Word |
| to compress with violence, out of natural shape or condition. | Squelch |
| a salt containing the anion $hco_3^-$. | Bicarbonate |
| the length of time that a person has lived or a thing has existed. | Age |
| used to indicate that one is waiting for an answer or explanation from someone. | Well |
| a quantity or supply of something kept for use as needed. | Store |
| a person or group that oppresses people. | Oppressor |

| | |
|---|---|
| eject the contents of the stomach through the mouth. | Spue |
| make a loud, high-pitched sound. | Scream |
| objective or physical; not subjective. | Outer |
| full of nervous energy, especially through taking amphetamines or similar drugs. | Amp |
| an adhesive solution; gum or glue. | Mucilage |
| a fastener consisting of two buttons joined with a bar, used in formal wear to fasten a shirt front or to fasten a collar to a shirt. | Stud |
| the air passage from the throat to the lungs; the trachea. | Windpipe |
| a curtain or piece of fabric fastened so as to hang in a drooping curve. | Swag |
| rope that is used for fastening something to something else. | Lashing |
| to say, state, or perform again. | Restate |
| being complete of its kind and without defect or blemish. | Perfect |
| creating a picture with paints. | Painting |
| make amorous advances towards. | Solicit |
| very beautiful or attractive. | Lovely |
| filled with soft feathers. | Downy |
| a high explosive consisting chiefly of a gel of nitroglycerine with added cellulose nitrate. | Gelatin |
| the capacity to experience the sense of touch. | Feeling |
| furnish with new or different furniture. | Refurnish |
| remove from the centre of activity or attention; place in a less influential position. | Sideline |
| rise up as in fear. | Uprise |
| the celebration of something in a joyful and exuberant way. | Festivity |
| stay or cause to stay at a certain value or level. | Hold |
| to arouse hope, desire, or curiosity without satisfying them. | Tease |
| liquid preparation having a soothing or antiseptic or medicinal action when applied to the skin. | Application |
| change or be different within limits. | Run |
| everything that exists anywhere. | Cosmos |
| uncomfortably humid or airless. | Close |
| a type of four-wheel-drive all-terrain military vehicle, or a similar vehicle intended for civilian use. | Hummer |
| covered with or containing or consisting of ice. | Icy |
| a caustic surface or curve. | Caustic |
| the antibody which is involved in allergic reactions, causing the release of histamine when it combines with antigen in tissue, and capable of producing sensitivity to the antigen when introduced into the skin of a normal individual. | Reagin |
| to prepare verbally, either for written or spoken delivery. | Prepare |
| a building or community occupied by or consisting of friars. | Friary |
| a preliminary round in a sporting competition. | Preliminary |
| load or cover with stacks. | Stack |
| a cavity in a plant, animal body, or organ. | Chamber |
| a periodic variation of an electromagnetic field in the propagation of light or other radiation through a medium or vacuum. | Wave |
| ornamentation by means of figures or designs. | Figuration |
| make or place parallel to something. | Collimate |
| be in accord; be in agreement. | Hold |
| brush or drive away with a waving movement. | Fan |
| vigorously energetic or forceful. | High-power |
| an australian acacia tree with delicate fern-like leaves and yellow flowers. | Mimosa |
| make hard or harder. | Harden |
| a tropical old world plant of the daisy family, with large brightly coloured flowers, cultivated under glass in cooler regions. | Gerbera |
| the round fruit of a tree of the rose family, which typically has thin green or red skin and crisp flesh. | Apple |

Table 5: 300 demonstrations used for in-context learning

## C  INFLUENCE OF QUANTIZATION

We analyze the influence of quantization in Table 6 between the 16bit models and 4bit models, which are quantized by bitsandbytes [1] with 4-bit normalfloat and double quantization. We find large models tend to show better results on STS tasks after 4-bit quantization. For example, PromptEOL+ICL with 6.7B OPT improve Spearman correlation from 79.08 to 79.38.

| Method | Params | STS12 | STS13 | STS14 | STS15 | STS16 | STS-B | SICK-R | Avg. |
|---|---|---|---|---|---|---|---|---|---|
| | 125M | 59.90 | 71.55 | 60.93 | 70.76 | 72.83 | 67.89 | 65.14 | 67.00 |
| | 350M | 54.70 | 71.52 | 59.99 | 64.51 | 71.39 | 66.55 | 66.58 | 65.03 |
| | 1.3B | 64.59 | 79.06 | 68.46 | 78.88 | 78.64 | 73.22 | 69.41 | 73.18 |
| PromptEOL | 2.7B | 60.03 | 75.51 | 64.30 | 74.56 | 77.62 | 67.73 | 65.35 | 69.30 |
| OPT(16-bit) | 6.7B | 60.91 | 80.05 | 67.65 | 75.49 | 80.11 | 72.91 | 67.57 | 72.10 |
| | 13B | 60.21 | 81.36 | 69.69 | 75.46 | 79.58 | 70.73 | 65.99 | 71.86 |
| | 30B | 59.99 | 80.52 | 69.80 | 75.20 | 78.03 | 73.57 | 69.87 | 72.43 |
| | 66B | 55.66 | 74.62 | 64.90 | 72.34 | 75.21 | 71.72 | 67.43 | 68.84 |
| | 125M | 60.53 | 70.03 | 59.02 | 69.77 | 72.38 | 66.47 | 65.17 | 66.20 |
| | 350M | 58.03 | 72.61 | 61.34 | 66.14 | 72.99 | 67.27 | 65.10 | 66.21 |
| | 1.3B | 63.72 | 79.32 | 68.13 | 77.92 | 78.56 | 72.03 | 68.80 | 72.64 |
| PromptEOL | 2.7B | 57.80 | 72.45 | 61.09 | 73.33 | 76.22 | 64.71 | 64.07 | 67.10 |
| OPT(4-bit) | 6.7B | 63.81 | 81.45 | 69.90 | 77.68 | 80.92 | 75.51 | 69.28 | 74.08 |
| | 13B | 60.91 | 80.97 | 70.22 | 76.93 | 79.46 | 72.84 | 66.34 | 72.52 |
| | 30B | 59.33 | 79.65 | 69.25 | 73.87 | 77.79 | 71.72 | 69.07 | 71.53 |
| | 66B | 59.35 | 77.33 | 68.33 | 74.45 | 77.25 | 73.93 | 69.27 | 71.42 |
| | 125M | 62.22 | 73.10 | 61.84 | 71.09 | 72.08 | 67.80 | 64.10 | 67.46 |
| | 350M | 63.87 | 73.85 | 63.41 | 72.45 | 73.13 | 70.84 | 65.61 | 69.02 |
| | 1.3B | 72.78 | 83.77 | 73.61 | 83.42 | 80.60 | 78.80 | 69.69 | 77.52 |
| PromptEOL+ICL | 2.7B | 68.49 | 84.72 | 75.15 | 83.62 | 81.34 | 80.94 | 72.97 | 78.18 |
| OPT(16-bit) | 6.7B | 70.65 | 84.51 | 75.01 | 83.51 | 82.00 | 81.12 | 76.77 | 79.08 |
| | 13B | 71.99 | 85.22 | 76.04 | 82.23 | 81.38 | 81.42 | 75.00 | 79.04 |
| | 30B | 69.99 | 83.35 | 74.75 | 83.14 | 82.42 | 81.45 | 77.46 | 78.94 |
| | 66B | 69.93 | 83.29 | 74.88 | 80.10 | 81.11 | 81.76 | 76.26 | 78.19 |
| | 125M | 61.02 | 71.00 | 59.75 | 69.67 | 70.52 | 65.14 | 63.45 | 65.79 |
| | 350M | 64.14 | 72.45 | 62.58 | 71.05 | 70.18 | 67.67 | 65.52 | 67.66 |
| | 1.3B | 73.45 | 82.55 | 73.11 | 83.63 | 80.60 | 78.72 | 69.06 | 77.30 |
| PromptEOL+ICL | 2.7B | 68.50 | 84.73 | 74.62 | 82.23 | 80.87 | 80.81 | 72.30 | 77.72 |
| OPT(4-bit) | 6.7B | 70.23 | 84.64 | 76.08 | 83.73 | 82.06 | 81.66 | 77.29 | 79.38 |
| | 13B | 71.79 | 84.23 | 75.57 | 81.75 | 80.71 | 80.89 | 74.46 | 78.49 |
| | 30B | 70.61 | 84.05 | 75.27 | 83.23 | 82.77 | 81.45 | 77.31 | 79.24 |
| | 66B | 71.67 | 83.95 | 75.67 | 81.33 | 81.86 | 82.58 | 76.54 | 79.09 |

Table 6: Influence of quantization on STS tasks. ICL denotes in-context learning with our demonstration set.

---

[1] https://github.com/TimDettmers/bitsandbytes

# D    TRANSFER TASKS

The results of PromptEOL with in-context learning (ICL) and contrastive learning (CSE) are shown in Table 7. Compared to PromptEOL, both PromptEOL+ICL and PromptEOL+CSE appeared to hinder performance on transfer tasks. We anticipate that the incorporation of additional datasets, such as the Community QA dataset, in accordance with ST5 Ni et al. (2021), or the implementation of full-model fine-tuning, might enhance the performance of PromptEOL+CSE in transfer tasks, which we leave in future. For PromptEOL+ICL, using STS-B or a dictionary as the example did not improve the performance on transfer tasks. We discover that using examples from a task with the label as the word in the example can improve the original performance. For instance, if we use one positive example and one negative example from training set of MR tasks, it increases the accuracy on MR in 6.7B OPT by approximately one point. We find these examples also beneficial to other transfer tasks, improving the average accuracy from 91.34 to 91.78, which can exceed 66B OPT performance.

| Method | Params | MR | CR | SUBJ | MPQA | SST | TREC | MRPC | Avg. |
|---|---|---|---|---|---|---|---|---|---|
| PromptEOL OPT | 125M | 80.86 | 87.66 | 93.19 | 89.77 | 87.31 | 92.20 | 72.64 | 86.23 |
| | 350M | 84.14 | 88.08 | 93.17 | 89.77 | 89.73 | 91.20 | 71.36 | 86.78 |
| | 1.3B | 88.06 | 91.55 | 95.90 | 91.55 | 93.08 | 95.00 | 73.97 | 89.87 |
| | 2.7B | 88.83 | 92.29 | 95.93 | 91.76 | 94.62 | 96.00 | 75.94 | 90.77 |
| | 6.7B | 90.26 | 92.50 | 96.67 | 91.39 | 94.67 | 96.00 | 77.91 | 91.34 |
| | 13B | 90.73 | 92.90 | 96.69 | 91.48 | 94.01 | 96.80 | 75.59 | 91.17 |
| | 30B | 90.95 | 92.77 | 96.99 | 91.79 | 95.28 | 97.00 | 73.97 | 91.25 |
| | 66B | 90.96 | 93.40 | 97.01 | 91.93 | 95.22 | 96.40 | 75.25 | 91.45 |
| PromptEOL+ICL OPT | 125M | 80.86 | 87.10 | 93.08 | 89.55 | 87.10 | 92.00 | 73.28 | 86.14 |
| | 350M | 82.20 | 86.65 | 93.21 | 89.70 | 87.86 | 87.60 | 72.52 | 85.68 |
| | 1.3B | 87.05 | 90.49 | 95.34 | 91.54 | 90.72 | 95.80 | 72.64 | 89.08 |
| | 2.7B | 88.73 | 91.79 | 95.44 | 91.54 | 93.52 | 95.20 | 75.30 | 90.22 |
| | 6.7B | 89.80 | 93.27 | 96.32 | 91.46 | 93.79 | 95.40 | 74.43 | 90.64 |
| | 13B | 89.45 | 92.98 | 96.23 | 91.28 | 94.51 | 95.40 | 75.71 | 90.79 |
| | 30B | 90.27 | 92.82 | 96.46 | 91.76 | 94.34 | 97.00 | 76.29 | 91.28 |
| | 66B | 90.40 | 92.50 | 97.08 | 91.24 | 94.34 | 97.40 | 75.01 | 91.14 |
| PromptEOL+CSE OPT | 1.3B | 88.62 | 91.89 | 95.49 | 91.64 | 94.29 | 94.80 | 73.22 | 89.99 |
| | 2.7B | 88.40 | 92.16 | 95.57 | 91.51 | 94.12 | 95.20 | 74.09 | 90.15 |
| | 6.7B | 89.60 | 92.05 | 95.91 | 91.09 | 94.78 | 95.80 | 75.71 | 90.71 |
| | 13B | 89.20 | 92.40 | 95.92 | 90.86 | 93.74 | 95.40 | 73.10 | 90.09 |
| PromptEOL LLaMA | 7B | 90.40 | 92.90 | 96.88 | 91.57 | 95.11 | 95.40 | 75.13 | 91.06 |
| | 13B | 92.02 | 93.22 | 97.29 | 91.40 | 95.66 | 95.80 | 76.46 | 91.69 |
| | 30B | 91.64 | 93.27 | 97.10 | 91.86 | 95.99 | 95.80 | 78.43 | 92.01 |
| | 65B | 92.13 | 93.43 | 97.16 | 91.91 | 95.33 | 97.40 | 77.28 | 92.09 |
| PromptEOL+CSE LLaMA | 7B | 90.28 | 93.27 | 96.67 | 91.45 | 94.73 | 95.60 | 75.54 | 91.08 |
| | 13B | 91.22 | 93.22 | 96.83 | 91.52 | 94.89 | 95.80 | 74.26 | 91.11 |

Table 7: Performances of our method with in-context learning and contrastive learning on transfer learning tasks.

# E    SENTENCE REPRESENTATION METHODS

We supplemented detail results in Table 8 and 9 for different sentence representation methods.

| Method | Params | STS12 | STS13 | STS14 | STS15 | STS16 | STS-B | SICK-R | Avg. |
|---|---|---|---|---|---|---|---|---|---|
| *Without fine-tuning* | | | | | | | | | |
| OPT avg. | 125M | 44.27 | 50.38 | 44.95 | 62.39 | 55.52 | 45.39 | 53.24 | 50.88 |
| | 350M | 40.61 | 47.25 | 40.45 | 55.12 | 55.57 | 40.53 | 47.66 | 46.74 |
| | 1.3B | 45.12 | 54.01 | 46.52 | 62.94 | 55.96 | 46.31 | 54.32 | 52.17 |
| | 2.7B | 44.11 | 54.35 | 47.89 | 63.91 | 57.02 | 47.85 | 54.44 | 52.80 |
| | 6.7B | 43.61 | 51.69 | 45.86 | 60.11 | 55.41 | 45.42 | 54.93 | 51.00 |
| | 13B | 46.95 | 54.92 | 48.74 | 60.13 | 54.96 | 48.07 | 53.93 | 52.53 |
| | 30B | 43.93 | 52.44 | 46.04 | 58.80 | 55.15 | 47.13 | 53.46 | 50.99 |
| | 66B | 40.81 | 47.98 | 44.21 | 59.37 | 56.37 | 43.80 | 53.19 | 49.39 |
| OPT prompt | 125M | 56.25 | 71.61 | 58.62 | 63.47 | 70.29 | 59.77 | 63.23 | 63.32 |
| | 350M | 56.56 | 69.27 | 55.81 | 60.05 | 68.73 | 61.75 | 64.15 | 62.33 |
| | 1.3B | 60.26 | 75.64 | 62.93 | 70.63 | 76.52 | 67.31 | 65.95 | 68.46 |
| | 2.7B | 59.34 | 75.47 | 62.64 | 69.76 | 75.65 | 68.35 | 67.48 | 68.38 |
| | 6.7B | 55.20 | 76.91 | 62.53 | 69.41 | 76.39 | 67.33 | 65.86 | 67.66 |
| | 13B | 49.60 | 75.43 | 61.58 | 67.33 | 75.53 | 65.98 | 63.79 | 65.61 |
| | 30B | 46.69 | 72.42 | 58.00 | 67.52 | 72.98 | 64.77 | 65.66 | 64.01 |
| | 66B | 50.21 | 69.65 | 56.78 | 70.20 | 73.37 | 64.31 | 66.93 | 64.49 |
| PromptEOL OPT | 125M | 59.90 | 71.55 | 60.93 | 70.76 | 72.83 | 67.89 | 65.14 | 67.00 |
| | 350M | 54.70 | 71.52 | 59.99 | 64.51 | 71.39 | 66.55 | 66.58 | 65.03 |
| | 1.3B | 64.59 | 79.06 | 68.46 | 78.88 | 78.64 | 73.22 | 69.41 | 73.18 |
| | 2.7B | 60.03 | 75.51 | 64.30 | 74.56 | 77.62 | 67.73 | 65.35 | 69.30 |
| | 6.7B | 60.91 | 80.05 | 67.65 | 75.49 | 80.11 | 72.91 | 67.57 | 72.10 |
| | 13B | 60.21 | 81.36 | 69.69 | 75.46 | 79.58 | 70.73 | 65.99 | 71.86 |
| | 30B | 59.99 | 80.52 | 69.80 | 75.20 | 78.03 | 73.57 | 69.87 | 72.43 |
| | 66B | 55.66 | 74.62 | 64.90 | 72.34 | 75.21 | 71.72 | 67.43 | 68.84 |
| *Fine-tuning on unsupervised datasets* | | | | | | | | | |
| OPT avg. | 125M | 74.08 | 82.70 | 77.76 | 83.65 | 79.74 | 82.43 | 78.55 | 79.84 |
| | 350M | 74.07 | 83.78 | 78.06 | 84.62 | 80.70 | 83.93 | 78.61 | 80.54 |
| | 1.3B | 75.38 | 84.99 | 80.34 | 86.10 | 81.49 | 84.35 | 79.98 | 81.80 |
| | 2.7B | 75.31 | 85.66 | 80.73 | 86.71 | 81.84 | 84.92 | 79.66 | 82.12 |
| | 6.7B | 76.02 | 86.22 | 81.30 | 87.07 | 82.54 | 85.28 | 80.53 | 82.71 |
| | 13B | 75.86 | 86.32 | 80.73 | 86.25 | 82.13 | 85.55 | 79.62 | 82.35 |
| OPT prompt | 125M | 76.05 | 85.24 | 79.82 | 85.27 | 81.30 | 84.56 | 79.09 | 81.62 |
| | 350M | 76.28 | 86.01 | 80.96 | 86.13 | 81.87 | 85.33 | 79.73 | 82.33 |
| | 1.3B | 78.56 | 89.21 | 84.21 | 88.71 | 84.17 | 87.39 | 81.16 | 84.77 |
| | 2.7B | 78.89 | 89.21 | 84.43 | 89.43 | 85.75 | 88.07 | 81.40 | 85.31 |
| | 6.7B | 78.66 | 89.81 | 84.45 | 89.70 | 85.71 | 88.63 | 81.79 | 85.54 |
| | 13B | 79.66 | 89.84 | 84.88 | 89.54 | 85.59 | 88.65 | 81.93 | 85.73 |
| PromptEOL OPT | 125M | 76.53 | 85.56 | 79.75 | 85.43 | 81.17 | 84.32 | 79.04 | 81.69 |
| | 350M | 75.96 | 85.51 | 81.32 | 86.50 | 81.42 | 85.24 | 80.35 | 82.33 |
| | 1.3B | 79.01 | 89.26 | 84.10 | 88.30 | 84.62 | 87.71 | 80.52 | 84.79 |
| | 2.7B | 79.49 | 89.64 | 84.80 | 89.51 | 85.91 | 88.33 | 81.64 | 85.62 |
| | 6.7B | 80.14 | 90.02 | 84.94 | 89.78 | 85.84 | 88.75 | 81.29 | 85.82 |
| | 13B | 80.20 | 90.24 | 85.34 | 89.52 | 85.90 | 88.56 | 82.06 | 85.97 |

Table 8: Comparison of three sentence representation methods on STS tasks.

| Method | Params | MR | CR | SUBJ | MPQA | SST | TREC | MRPC | Avg. |
|---|---|---|---|---|---|---|---|---|---|
| OPT avg. | 125M | 80.63 | 86.41 | 93.91 | 87.85 | 86.22 | 92.60 | 71.83 | 85.64 |
| | 350M | 80.73 | 85.16 | 93.42 | 87.26 | 86.11 | 87.80 | 69.57 | 84.29 |
| | 1.3B | 85.89 | 90.04 | 95.71 | 90.10 | 91.38 | 94.20 | 72.99 | 88.62 |
| | 2.7B | 87.55 | 90.76 | 95.78 | 90.26 | 91.71 | 94.40 | 68.00 | 88.35 |
| | 6.7B | 87.93 | 91.07 | 96.58 | 90.65 | 92.70 | 96.20 | 72.17 | 89.61 |
| | 13B | 88.33 | 91.76 | 96.74 | 90.78 | 93.25 | 95.20 | 70.90 | 89.57 |
| | 30B | 88.54 | 92.11 | 96.85 | 90.61 | 93.74 | 94.40 | 70.72 | 89.57 |
| | 66B | 89.17 | 92.00 | 96.86 | 90.80 | 94.67 | 96.40 | 71.07 | 90.14 |
| OPT prompt | 125M | 83.54 | 87.60 | 94.28 | 89.36 | 88.74 | 91.60 | 67.01 | 86.02 |
| | 350M | 80.99 | 84.08 | 93.30 | 89.38 | 86.88 | 88.80 | 60.99 | 83.49 |
| | 1.3B | 87.31 | 90.68 | 95.73 | 91.30 | 93.47 | 94.40 | 72.99 | 89.41 |
| | 2.7B | 88.58 | 91.60 | 96.22 | 91.36 | 93.90 | 95.80 | 70.96 | 89.77 |
| | 6.7B | 90.55 | 92.21 | 97.09 | 91.31 | 95.06 | 96.60 | 74.90 | 91.10 |
| | 13B | 90.45 | 92.66 | 96.85 | 91.57 | 95.44 | 96.00 | 74.55 | 91.07 |
| | 30B | 90.56 | 92.79 | 97.28 | 91.93 | 94.78 | 96.00 | 72.93 | 90.90 |
| | 66B | 90.95 | 92.48 | 97.27 | 91.72 | 95.55 | 95.80 | 75.30 | 91.30 |
| PromptEOL OPT | 125M | 80.86 | 87.66 | 93.19 | 89.77 | 87.31 | 92.20 | 72.64 | 86.23 |
| | 350M | 84.14 | 88.08 | 93.17 | 89.77 | 89.73 | 91.20 | 71.36 | 86.78 |
| | 1.3B | 88.06 | 91.55 | 95.90 | 91.55 | 93.08 | 95.00 | 73.97 | 89.87 |
| | 2.7B | 88.83 | 92.29 | 95.93 | 91.76 | 94.62 | 96.00 | 75.94 | 90.77 |
| | 6.7B | 90.26 | 92.50 | 96.67 | 91.39 | 94.67 | 96.00 | 77.91 | 91.34 |
| | 13B | 90.73 | 92.90 | 96.69 | 91.48 | 94.01 | 96.80 | 75.59 | 91.17 |
| | 30B | 90.95 | 92.77 | 96.99 | 91.79 | 95.28 | 97.00 | 73.97 | 91.25 |
| | 66B | 90.96 | 93.40 | 97.01 | 91.93 | 95.22 | 96.40 | 75.25 | 91.45 |

Table 9: Comparison of three sentence representation methods on STS tasks.

## F   RESULST OF PROMPTEOL AND PROMPTEOL+ICL ON CURRENT POPULAR LLMS

We supplemented results of STS tasks with PromptEOL and PromptEOL+ICL in Table 10 on current popular LLMs include open-LLaMA Geng & Liu (2023), LLaMA Touvron et al. (2023a), LLaMA-2 Touvron et al. (2023b), MPT MosaicML (2023), Mistral Jiang et al. (2023).

| Params | Avg. | Prompt | PromptEOL | PromptEOL+ICL |
|--------|------|--------|-----------|---------------|
| *Open-LLaMA* | | | | |
| 3B | 51.75 | 66.45 | 68.22 | 78.85 |
| 7B | 52.03 | 63.40 | 76.35 | 79.17 |
| 13B | 49.58 | 64.11 | 70.03 | 78.04 |
| *LLaMA* | | | | |
| 7B | 46.94 | 42.18 | 68.76 | 77.63 |
| 13B | 47.53 | 48.73 | 65.62 | 73.40 |
| 30B | 50.70 | 47.10 | 70.60 | 77.61 |
| 65B | 44.80 | 51.69 | 69.39 | 75.73 |
| *LLaMA-2* | | | | |
| 7B | 46.34 | 45.87 | 69.30 | 75.99 |
| 13B | 49.07 | 58.80 | 68.87 | 78.31 |
| 70B | 44.34 | 45.14 | 70.90 | 74.97 |
| *MPT* | | | | |
| 7B | 49.39 | 57.25 | 71.06 | 79.08 |
| 30B | 42.31 | 54.45 | 71.08 | 75.74 |
| *Mistral* | | | | |
| 7B | 49.32 | 66.23 | 73.32 | 78.35 |

Table 10: Results of PromptEOL and PromptEOL+ICL on current popular LLMs. We report averaging Spearman correlation over seven STS tasks with four sentence representation methods: avg., prompt, PromptEOL and PromptEOL+ICL. "Avg." refers to use averaging output tokens as sentence embeddings. "Prompt" refers to extract sentencne embeddings using the template from Jiang et al. (2022). For simplicity, we do not search demonstration for PromptEOL+ICL but use the best demonstration from the PromptEOL+ICL OPT directly. We expect that PromptEOL+ICL can achieve better results by searching for demonstrations according to the model.

