# OpenReview forum: "Scaling Sentence Embeddings with Large Language Models"
_ICLR.cc/2024/Conference — Submitted to ICLR 2024_

### Official Review · Reviewer_ViaW · 2023-10-29

**Soundness:** 3 good
**Presentation:** 2 fair
**Contribution:** 3 good
**Rating:** 6
**Confidence:** 4

**Summary:**

This paper works on generating sentence embeddings using large language models. First,  the authors design a specific prompt  to compress the semantic of an input sentence into a single word.  Then, this paper investigates zero-shot,  in-context and fine-tuning settings of sentence embedding learning.  For in-context learning,  this paper proposes a demonstration selection method for inducing good sentence representations.  For fine-tuning, to solve the large memory issue,  the authors use QLoRA to perform contrastive learning.  Empirical results on common sentence embedding evaluation benchmarks with both OPT and LLaMA series models show that the proposed method can match (or even exceed) the performance of pretrained language models (such as BERT).

**Strengths:**

1. The writing is easy to follow and the idea is well presented.
2. The proposed prompt, in-context demonstration and fine-tuning method solve the specific issues of scaling large language models for sentence embedding learning.
3. The experimental results are effective on both the SentEval and Transfer settings compared to BERT-base based contrastive learning method.

**Weaknesses:**

1. In Table 1,  only the results based on OPT are presented.  Why not also including the results based on LLaMA?
2. In Table 1,  the best configuration PromptEOL+ICL+ OPT (6.7B) does not show clear advantages than PromptRoBERTa (123M).
3. For the in-context setting,  why only use one demonstration?   In Table 1,  comparing PromptEOL+ICL + OPT with baselines models  is not fair since the baseline models do not use the development set.
4. When the model size increases, the performance does always not increase.  Especially, the 13B, 30B, and 60B models do not perform better than smaller models such as 1.3B and 6B models.
5. The overall method is a little bit heavy. It is worth to discuss whether we should improve the sentence embeddings using large language models.

**Questions:**

1. Do you also try LLaMA 2?
2. In Equation 1, why using the last token hidden state as the sentence representation instead of the representation vector of the last generated token using the explicit one word prompt?
3. For the fine-tuning, do you also try including in-context demonstration for the fine-tuning?

Minors:

The citation format is not correct. Please correct all of them.  Try to use the cite command in a correct way.

---

> ### Author Response · Authors · 2023-11-16
> **Response to Reviewer ViaW (Part 1/2)**
>
> We sincerely thank you for your helpful feedback and insightful comments. We address your comments and questions below.
>
> > **W1:** In Table 1, only the results based on OPT are presented. Why not also including the results based on LLaMA?
>
> In this work, the main focus is on scaling up sentence embeddings with various model sizes. While LLaMA only has four model sizes, we use OPT as the primary model to showcase the results.
>
> To address the concern about the robustness of our method, we evaluated the performance of PromptEOL and PromptEOL+ICL on current popular LLMs, varying in size from 3B to 70B. The evaluated LLMs include LLaMA, LLaMA-2, open-LLaMA, MPT, and Mistral.
> Our method shows significant improvement across above 13 models. PromptEOL achieves an average Spearman correlation score of 70.27 on STS tasks among the above models, while PromptEOL+ICL achieves an average score of 77.14. Comparing to the average pooling baseline, PromptEOL and PromptEOL+ICL achieve average improvement of 22.27 and 29.14, respectively. Furthermore, when compared to the previous prompt representation method, PromptEOL and PromptEOL+ICL exhibit an average improvement of 15.55 and 22.42. We have added results in Appendix F of our paper.
>
>
> > **W2:** In Table 1, the best configuration PromptEOL+ICL+ OPT (6.7B) does not show clear advantages than PromptRoBERTa (123M)
>
> PromptRoBERTa is fine-tuned using contrastive learning, whereas PromptEOL+ICL does not require any fine-tuning. It may not be fair to directly compare these two methods. However, PromptEOL+ICL also achieves similar performance to PromptRoBERTa, which demonstrates that a general language model can achieve such performance without relying on contrastive learning or fine-tuning.
> We also introduce other advantages benefit from PromptEOL+ICL in **W5**.
>
> > **W3:** For the in-context setting, why only use one demonstration? In Table 1, comparing PromptEOL+ICL + OPT with baselines models is not fair since the baseline models do not use the development set.
>
> We found that one demonstration is sufficient for LLMs to generate good sentence embeddings. Increasing the number of demonstrations does not further improve the performance.
> We believe the comparison is fair. The ICL examples used in PromptEOL+ICL are taken from STS-B training set and Oxford dictionary. We only use development set to evaluate the performance of ICL examples. In comparison, other baselines such as SimCSE and PromptBERT also utilize STS-B development set to select the best checkpoint during training. Moreover, BERT prompt also utilizes the STS-B development set for prompt searching.
>
> > **W4:** When the model size increases, the performance does always not increase. Especially, the 13B, 30B, and 60B models do not perform better than smaller models such as 1.3B and 6B models.
>
> In this work, we aim to investigate the capability of LLMs on sentence embeddings by scaling up the model size. As we mention in our paper, we find that continuing to scale up to over ten billion parameters does not always improve the performance of sentence embeddings for STS tasks. We believe that our work can provide valuable insights for future research on leveraging LLMs for sentence embeddings.
>
> > **W5:** The overall method is a little bit heavy. It is worth to discuss whether we should improve the sentence embeddings using large language models.
>
> First, our work not only works for LLMs (7B, 13B, 60B models), but also smaller models like OPT 1.3B or 2.7B. Furthermore, we significantly reduce the computation cost of fine-tuning high-quality sentence embeddings compared to the previous SOTA methods like ST5. We can exceed the previous 4.8B ST5 with 2.7B OPT fine-tuned on less training data. Compared to 4.8B ST5,  it uses two-stage training with 2 billion extra question-answers pairs and 275k NLI datasets by fine-tuning all parameters, PromptEOL+CSE can leverage efficient fine-tuning method QLoRA on 275k NLI datasets with one epoch. We can train 2.7B OPT with PromptEOL+CSE for around one hour on four 3090 GPUs. However, it will take more than hundreds of hours with the same hardware environment to train 4.8B ST5.
>
> Second, PromptEOL+ICL also provides a lightweight method that allows LLMs to generate sentence embeddings without the need of fine-tuning. We can use LLMs themselves to embedding sentence to perform retrieval augmented generation, without additional embedding models. Moreover, we can control the embedding behaviors by modifying prompt and ICL example to suit specific application, such as cross-lingual representation. For instance, we can utilize an ICL example with a Chinese sentence and an English summary word to project the semantics of Chinese sentences onto an English word. This can help bridge the language gap in sentence embeddings, which we leave for future work.

---

> > ### Author Response · Authors · 2023-11-16
> > **Response to Reviewer ViaW (Part 2/2)**
> >
> > > **Q1:** Do you also try LLaMA 2?
> >
> > Yes, we have tried LLaMA-2 with PromptEOL+CSE, it achieves slightly better results compared to LLaMA.
> >
> > |                           | STS12 | STS13 | STS14 | STS15 | STS16 | STS-B | SICK-R |  Avg. |
> > | -------                   |  :--: |  :--: |  :--: |  :--: |  :--: |  :--: |   :--: |  :--: |
> > | PromptEOL+CSE LLaMA 7b    | 79.16 | 90.22 | 85.40 | 88.99 | 86.25 | 88.37 |  81.51 | 85.70 |
> > | PromptEOL+CSE LLaMA 13b   | 78.63 | 90.03 | 85.46 | 89.48 | 86.18 | 88.45 |  82.69 | 85.85 |
> > | PromptEOL+CSE LLaMA-2 7b  | 78.48 | 90.07 | 84.86 | 89.43 | 86.16 | 88.44 |  83.20 | 85.81 |
> > | PromptEOL+CSE LLaMA-2 13b | 78.84 | 90.35 | 85.88 | 89.72 | 86.68 | 88.91 |  82.64 | 86.15 |
> >
> > > **Q2:** In Equation 1, why using the last token hidden state as the sentence representation instead of the representation vector of the last generated token using the explicit one word prompt?
> >
> > The last token hidden state is utilized to predict the next token in an autoregressive model. When using the explicit one-word prompt, the last token hidden state can capture the semantic meaning of the sentence.
> >
> > For the last generated token, there are several problems with it. First, since LLMs are not instruction tuning, the generation can be uncontrollable. For example, LLMs may generate additional tokens even after providing a one-word summary of the sentence. Second, due to LLMs predicting token one by one, the generation process is slower compared to using the last token hidden state.
> >
> > > **Q3:** For the fine-tuning, do you also try including in-context demonstration for the fine-tuning?
> >
> > Yes, we did try including in-context demonstrations for the fine-tuning. However, it did not help improve the fine-tuning performance. For example, on OPT 6.7B, it achieved an average Spearman correlation of 85.56 on STS tasks, compared to the original results of 85.82.

---

> > > ### Author Response · Authors · 2023-11-21
> > > **Does our response address your concerns?**
> > >
> > > Dear reviewer ViaW,
> > >
> > > Thanks very much for your time and effort. As the discussion phase is closing, could you review our responses to ensure they have addressed your concerns? We would also appreciate any additional feedback you might have for improvements.
> > >
> > > Best regards
> > >
> > > Authors of 596

---

> > > > ### Comment · Reviewer_ViaW · 2023-11-21
> > > >
> > > > Thanks for the reminder and the rebuttal.   I can increase my score to 7 but since there is no such an option, I just write it down to  let the AC and the other reviewers know.
> > > >
> > > > For the rebuttals,  I can fully accept W1 and W3.  Especially thanks for the clarifications of W3.
> > > >
> > > > The reasons I did not improve my score to 8 are based on two points.
> > > >
> > > > First,  at the high-level, it is not always easy to compress a sentence into a single word.  It can work to some extent for short sentences but the limitation is also very heavy for long sentences.
> > > >
> > > > Second,  the model performances do not well scale up for model sizes.  This raises concerns why we need large language models for sentence embedding models.  If we aim for small embedding models,  fully finetuning using contrastive learning (can be assisted with in-context examples) is not so expensive, which still can be a good option.  But if we aim for large embedding models, the performances do not improve, which might not be cost-effective.  These concerns put the proposed method in an little embarrassing position.
> > > >
> > > > In conclusion,  I believe the proposed method is novel and this is first work in this direction.  However, whether the designed method can fully activate the abilities of LLMs for sentence embedding models is still worth to discuss.

---

> ### Author Response · Authors · 2023-11-21
>
> Thanks for responses. We greatly appreciate your positive comments.
>
> However, we would like to clarify above points:
>
> * First, our method does not compress a sentence directly into a single word, but rather uses hidden states as sentence embeddings. This reflects the probability of words that best summarize the sentence, rather than a single word. This ensures that our method also works on long sentence tasks like SUBJ in transfer tasks.
>
> * Second, scaling up works well on transfer tasks and STS tasks with fine-tuning (PromptEOL+CSE). We noticed that the model performance doesn't scale up as effectively only on STS tasks without fine-tuning (PromptEOL+ICL). We also discussed the reason for this issue in **Q1** for reviewer [dT5b](https://openreview.net/forum?id=V0CUOBWUHa&noteId=0USsLQ9H9a).
> Regarding the performance concerns of our method:
>   * Our method with LLMs achieves the state-of-the-art results on transfer tasks and STS tasks, demonstrating the potential of using LLMs to generate sentence embeddings. Our method significantly improves the performance over small models on Table 2 and 3.
>   * Since contrastive learning can alleviate the anisotropy problem of sentence embeddings[1], comparing a fine-tuned model with PromptEOL+ICL might seem unfair. Nonetheless, PromptEOL+ICL still achieves comparable performance. When considering baselines without fine-tuning, LLMs can benefit from in-context learning with PromptEOL+ICL, showing strong advantages over other methods. We also discuss the benefits of in-context learning in the second point of **W5**.
>   * We provide PromptEOL+CSE for the fine-tuning setting, which outperforms small models by more than 3 points on the Spearman correlation, even with limited computational resources. For instance, we can efficiently fine-tune a 2.7B OPT with PromptEOL+CSE for approximately one hour on four 3090 GPUs.
>
> Thank you again for your time and effort. Please let us know if you have further questions.
>
> **Reference**
>
> [1] Gao T, Yao X, Chen D. SimCSE: Simple Contrastive Learning of Sentence Embeddings (EMNLP 2021)

---

> ### Author Response · Authors · 2023-11-23
> **Waiting for further discussion**
>
> Dear reviewer ViaW,
>
> Thanks very much for your responses. As the discussion phase is going to end today, could you review our recent responses adequately addressed your concerns?
>
>  We sincerely thank you for your positive feedback.
>
> Best regards
>
> Authors of 596

---

> ### Comment · Reviewer_ViaW · 2023-11-23
>
> Your prompt is ```This sentence: “xi” means in one word:```, which intends to compress the whole semantics into a single word. The hidden state vectors represent the intention of your prompt.  I do not know why you are now claiming you are not compressing it.   Ideally, this prompt will compress any input sentences into a class in the output vocab, which might be too aggressive.
>
> SUBJ is a simple task for identifying subjectivities of user's movie review.  The performance on SUBJ can not directly and fully show the embedding quality.
>
> For scale,  I mean your 66B model give much lower performance than your 1.3B model in Table 1 for the PromptEOL
> OPT setting. Similarly, for the PromptEOL+ICL+OPT setting,  the 66B model does not show clear advantage of the 13B model.
>
> I have clearly expressed my score will be 7. And this is my final score.  Thanks.

---

> > ### Author Response · Authors · 2023-11-23
> >
> > Thanks very much for your responses. It seems there's some misunderstanding about our clarifications.
> >
> > We don't compress the sentence into a class in the output vocab. Instead, we use the last token hidden state from the last layer as the sentence embedding, which is used to predict the probabilities for all classes in the output vocab in LLMs.
> > Like PromptBERT[1],  the prompt is used to instruct model to generate a better sentence representation, not to actually generate a single word or a class in the output vocab. For the concerns about the expressiveness of our sentence method, our method outperforms other representation methods by more than 10 points in all 14 sentence embedding tasks with current popular LLMs.
> >
> > For scaling, we attribute the reason for the failure of the scaling law on STS tasks without fine-tuning to the anisotropy of sentence embeddings[2]. We have noticed that as the model size increases, the anisotropy of sentence embeddings also increases. Although larger models may have a higher overall capacity, the issue of anisotropy can limit their performance. However, if we employ contrastive learning to mitigate anisotropy, scaling up can still be effective in Table 2.
> > For the concern about the performance of LLMs, we demonstrate that LLMs have significant potential for generating high-quality sentence embeddings.
> >
> > We sincerely thank you for your positive feedback.
> >
> > **Reference**
> >
> > [1] Jiang T, Jiao J, Huang S, et al. PromptBERT: Improving BERT Sentence Embeddings with Prompts (EMNLP 2022)
> >
> > [2]Gao T, Yao X, Chen D. SimCSE: Simple Contrastive Learning of Sentence Embeddings (EMNLP 2021)

---

### Official Review · Reviewer_oEkP · 2023-10-30

**Soundness:** 3 good
**Presentation:** 3 good
**Contribution:** 3 good
**Rating:** 5
**Confidence:** 3

**Summary:**

This paper proposes a set of methods leveraging LLMs for sentence embeddings.

* It introduces a prompting strategy with explicit one word limitation, pushing the model to condense as much information as possible into the last hidden representation. This method is an adaptation of PromptBERT's approach for autoregression models.
* It leverages in-context-learning in order to improve the quality of sentence embeddings. To this end, it relies on two approaches: 1) It generates one-word summaries of sentences from the STS training set using GPT-3.5. 2) It leverages entries from the Oxford dictionary. The concatenation of samples from these two sources is then incorporated into the LLM's prompt.
* It leverages fine-tuning with contrastive learning to further improve the quality of sentence embeddings. It does so by leveraging qLORA and training on supervised datasets such as SNLI and MNLI.


The paper's conclusions are as follow:
* The explicit one-word limitation prompt improves the quality of sentence embeddings derived from OPT on STS benchmarks.
* In-context learning and supervised fine-tuning improve the performance on STS benchmarks, allowing the proposed solution to beat the state of the art. However, the resulting embeddings do not transfer as well to other tasks.
* In the paper's proposed setup, the largest base models do not have a clear performance advantage: best results on STS without fine-tuning are obtained with OPT's variants ranging between 1.3 and 6.7B parameters.

**Strengths:**

* Originality: the proposed PromptEOL is a novel adaptation of the BERT prompting paradigm for sentence representations. The prompting strategy combining GPT-augmented STS sentences and oxford definitions is novel as well, and the use of qLORA to make contrastive fine-tuning feasible for larger models shows creativity in putting together existing solutions.

* Quality: The experiments are well-devised and executed. The proposed methods are simple and beat the state of the art on semantic textual similarity benchmarks.

* Clarity: The paper articulates very clearly its methodology. It is easy to read and describes well the corresponding pre-existing work. It motivates very well the choice of an explicit one-word prompt, the value of in-context learning and the need for quantization in order to fine-tune the largest models in a contrastive learning setup.

* Significance: while the results on STS benchmarks look good, both in-context learning and contrastive fine-tuning do not show incremental value on transfer tasks. The relatively low generalisation capabilities of these methods limit greatly the appeal of such techniques for the average practitioner, as most real-life applications of sentence embeddings are not for semantic textual similarity.

**Weaknesses:**

* While it is helpful to the reader to see the entire distribution of Spearman correlations, it may be relevant to give more details on how the two sources for ICL data impact the quality of downstream representations. The 1/3-2/3 mix of STS sentences vs Oxford definitions would benefit from an explicit ablation.

* The value of ICL and CSE is demonstrated only for OPT. Indeed, table 1 is missing results that would demonstrate the added value of ICL and CSE on Llama.

* The proposed methods (in-context learning and possibly fine-tuning) are performing worse than simple explicit-one-word-limit prompting on transfer tasks.

* It is not clear what section 5.2 demonstrates:
  * first, the text mentions "in-context learning examples that were obtained from each model on the STS-B development set", while the table caption reads "In-context learning examples used in various model size". The paper states clearly in section 3.2 that the in-context learning examples (1) come from the STS-B training set and (2) are not generated / obtained from the model itself.
  * second, the method used to sample the data from table 4 is not described, and the meaning of the "Improve" column is not clear: does it correspond to the improvement coming from one additional sentence in the prompt? Or from the addition of the 100s of sentences in the ICL prompt? In any case, it seems premature to draw generic conclusions such as "related examples are usually more implicit" from a sample size of 1 from each model. Appending 5-10 random samples of each category in the appendix would give more compelling evidence for this.
  * finally, it is not clear how to relate the findings of section 5.2 to the overall quality of the sentence embeddings introduced by this work.

* Typos and errors:
  * table 6, the first group of data rows should mention (16-bit)
  * table 8, the ordering is not the same between the "Without fine-tuning" and "Fine-tuning on unsupervised datasets" groups of rows.

**Questions:**

1. It would seem that some experiments have been run only on OPT, while others have been run on OPT and Llama. It would be helpful to have all experiments run on both models to show that the conclusions are robust to the choice of base LLM.

2. How was the data mix for ICL (1/3 STS sentences, 2/3 Oxford definitions) devised? Are they both necessary to achieve good performance? A proper ablation of this setup would be helpful.

---

> ### Author Response · Authors · 2023-11-16
> **Response to Reviewer oEkP (Part 1/2)**
>
> We sincerely thank you for your helpful feedback and insightful comments. We address your comments and questions below.
>
> > **W1:** While it is helpful to the reader to see the entire distribution of Spearman correlations, it may be relevant to give more details on how the two sources for ICL data impact the quality of downstream representations. The 1/3-2/3 mix of STS sentences vs Oxford definitions would benefit from an explicit ablation.
>
> We would like to clarify that in our method, only one example from the ICL data is used to perform sentence embeddings. To select this example, we evaluate the performance of each example on the STS-B development set and choose the one that performs the best. This process is described in detail in Section 3.2 of our paper. The proportion of STS-B data and the Oxford dictionary in our method is not a crucial factor; we use a 1/3-2/3 mix simply because obtaining data from the Oxford dictionary is much easier compared to generating words from ChatGPT.
>
> > **W2:** The value of ICL and CSE is demonstrated only for OPT. Indeed, table 1 is missing results that would demonstrate the added value of ICL and CSE on Llama.
>
> In this work, we focus on scaling up sentence embeddings with various model sizes. Compared to OPT, LLaMA only has four model sizes. We use OPT as the main model to show the results. Due to the page limit, we only present the ICL results on OPT. However, we have included the CSE on LLaMA in Table 2.
>
> To address the concern about the robustness of our method, we evaluated the performance of PromptEOL and PromptEOL+ICL on current popular LLMs, varying in size from 3B to 70B. The evaluated LLMs include LLaMA, LLaMA-2, open-LLaMA, MPT, and Mistral.
> Our method shows significant improvement across above 13 models. PromptEOL achieves an average Spearman correlation score of 70.27 on STS tasks among the above models, while PromptEOL+ICL achieves an average score of 77.14. Comparing to the average pooling baseline, PromptEOL and PromptEOL+ICL achieve average improvement of 22.27 and 29.14, respectively. Furthermore, when compared to the previous prompt representation method, PromptEOL and PromptEOL+ICL exhibit an average improvement of 15.55 and 22.42. We have included detailed results in Appendix F of our paper.
>
> > **W3:** The proposed methods (in-context learning and possibly fine-tuning) are performing worse than simple explicit-one-word-limit prompting on transfer tasks.
>
> Benefiting from our PromptEOL method, LLMs can capture the semantics of sentences well for transfer tasks. In Figure 4(c), we demonstrate that PromptEOL significantly outperforms other representation methods like prompt and average pooling on transfer tasks, achieving state-of-the-art results without fine-tuning.
> For in-context learning, we can flexibility modify the prompt to generate embeddings that fit the task following [1]. For example, by using a sentiment classification example, we can improve the results of PromptEOL 6.7B OPT from 91.34 to 91.78 average accuracy on seven transfer tasks.
> For fine-tuning, due to limited computational resources, we only use efficient fine-tuning LLMs with QLoRA, which may harm the original transfer task performance through 4-bit quantization.
>
> **References:**
>
> [1] Su H, Kasai J, Wang Y, et al. One embedder, any task: Instruction-finetuned text embeddings (ACL findings 2023)

---

> > ### Author Response · Authors · 2023-11-16
> > **Response to Reviewer oEkP (Part 2/2)**
> >
> > > **W4:** It is not clear what section 5.2 demonstrates...
> >
> > We would like to clarify the misunderstandings regarding section 5.2 and our PromptEOL+ICL method.
> > * The text mentioning "in-context learning examples that were obtained from each model on the STS-B development set" is reference to the best performance example on STS-B development set as introduced in section 3.2. This example comes from STS-B training set and Oxford dictionary.
> >
> > * The method used to sample the data is introduced in section 3.2, which we evaluate the performance of each example on STS-B development set and choose the best one. The "Improve" column is the average improvement of the PromptEOl+ICL compared to PromptEOL in the seven STS test set.
> >
> > * For the conclusion that "related examples are usually more implicit," it is important to note that the samples in Table 4 were not randomly selected from the ICL examples. Instead, they were chosen based on the performance of the corresponding model on the STS-B development set. In other words, the samples listed in Table 4 represent the most suitable example among 300 ICL examples for each model to represent sentences as embeddings. For "appending 5-10 random samples of each category", sentence embeddings, as a task to represent the sentences as vectors, do not have categories like the classification task.
> >
> > * For the relationship between section 5.2 and our methods, section 5.2 is to report that the ICL examples used in PromptEOL+ICL. It also show the improvement of PromptEOL+ICL compared to PromptEOL in the "Improve" column.
> >
> > > **Q1:** It would seem that some experiments have been run only on OPT, while others have been run on OPT and Llama. It would be helpful to have all experiments run on both models to show that the conclusions are robust to the choice of base LLM.
> >
> > Please refer to **W2**.
> >
> > > **Q2:** How was the data mix for ICL (1/3 STS sentences, 2/3 Oxford definitions) devised? Are they both necessary to achieve good performance? A proper ablation of this setup would be helpful.
> >
> > Please refer to **W1** and **W4**.

---

> ### Author Response · Authors · 2023-11-23
> **Does our response address your concerns?**
>
> Dear reviewer oEkP,
>
> Thanks very much for your time and effort. As the discussion phase is going to end today, could you review our responses to ensure they have addressed your concerns? We would also appreciate any additional feedback you might have for improvements.
>
> Best regards,
>
> Authors of 596

---

### Official Review · Reviewer_dT5b · 2023-10-31

**Soundness:** 3 good
**Presentation:** 3 good
**Contribution:** 2 fair
**Rating:** 6
**Confidence:** 5

**Summary:**

This paper investigates how to better leverage large language models (LLMs) for generating sentence representations, traditionally obtained from smaller encoder-based models like BERT variants.
It introduces two approaches.
Initially, it adopts in-context learning, similar to the utilization of LLMs in other tasks.
Employing the "Explicit One word Limitation (EOL)"—which posits that decoder-based models can produce viable sentence-level representations when prompted to summarize a sentence in a single word—sentence-to-word pair contexts are used to enhance representation derivation.
Decoder models, without any fine-tuning, showed performance on par with existing contrastive learning approaches.
Additionally, the authors explored fine-tuning decoder models using the prevalent contrastive learning framework in sentence representation research, employing the parameter-efficient technique known as QLoRA.
The findings reveal that fine-tuning with contrastive learning notably benefits larger decoder models, surpassing smaller encoder models in both Semantic Textual Similarity (STS) benchmarks and transfer tasks for classification.

**Strengths:**

- Suggested a variety of plausible methods for utilizing Large Language Models (LLMs) to compute sentence representations.
- Explored both in-context learning and fine-tuning approaches with LLMs, encompassing a broad spectrum of potential applications for these models.
- Introduced a straightforward yet insightful technique for integrating in-context learning into the sentence representation learning paradigm.

**Weaknesses:**

- While the methods proposed are sound, they consist of previously suggested and widely implemented techniques, which diminishes the novelty aspect of the work.
- Contrary to SimCSE, the in-context learning approach depends on the use of the STS-B dataset, including its training and validation components, which could potentially confer an unfair advantage to the method.
- There appears to be no direct link between the two proposed methods; that is, the approach based on in-context learning and the one utilizing contrastive learning.

**Questions:**

- I'm curious whether the authors have any insights or hypotheses as to why (much) larger models (over 10B) do not excel as expected in computing sentence representations, which contrasts with their effectiveness in other standard applications.

---

> ### Author Response · Authors · 2023-11-16
> **Response to Reviewer dT5b**
>
> We sincerely thank you for your helpful feedback and insightful comments. We address your comments and questions below.
>
> > **W1:** While the methods proposed are sound, they consist of previously suggested and widely implemented techniques, which diminishes the novelty aspect of the work.
>
> The primary focus of our research is the scaling up of sentence embeddings using LLMs, which remains unexplored. We also proposed three methods to leverage the capacity of LLMs for sentence embeddings: PromptEOL, PromptEOL+ICL, and PromptEOL+CSE.
>
> While our work is inspired by earlier studies such as PromptBERT, we have identified that PromptBERT does not extend its applicability to LLMs for sentence embeddings. To address this shortcoming, we have proposed a solution named PromptEOL. This method introduces an implicit single-word limitation to the prompt, thereby facilitating the generation of sentence embeddings. It is necessary to mention that, as the output of sentence embeddings results in a singular vector, in-context learning cannot be applied directly. To overcome this challenge, we propose PromptEOL+ICL. To the best of our knowledge, we are the first to show that in-context learning can be applied to sentence embeddings without fine-tuning.
>
> > **W2:** Contrary to SimCSE, the in-context learning approach depends on the use of the STS-B dataset, including its training and validation components, which could potentially confer an unfair advantage to the method.
>
> We think the comparison with SimCSE and other similar methods is fair. First, SimCSE also uses STS-B validation set for selecting the best checkpoint during training. Second, we do not use any label information from the STS-B training set. Instead, we only utilize 100 sentences from the STS-B training set, ensuring that none of these sentences come from the same sentence pairs. In fact, our approach outperforms SimCSE even when solely leveraging the Oxford dictionary. Finally, our ICL method does not rely on fine-tuning or contrastive learning, which are considered essential for achieving better representation in current sentence embeddings methods[1,2].
>
> > **W3:** There appears to be no direct link between the two proposed methods; that is, the approach based on in-context learning and the one utilizing contrastive learning.
>
> Both in-context learning and contrastive learning methods are built upon our proposed representation method, namely PromptEOL. This method focuses on two settings in sentence embeddings: with and without fine-tuning. In the case of in-context learning, we demonstrate that combining PromptEOL with in-context learning yields high-quality sentence representations without the need for fine-tuning. Additionally, in the context of contrastive learning, we provide evidence in Figure 4(b) that PromptEOL can benefit from contrastive learning, outperforming other representation methods.
>
> > **Q1:** I'm curious whether the authors have any insights or hypotheses as to why (much) larger models (over 10B) do not excel as expected in computing sentence representations, which contrasts with their effectiveness in other standard applications.
>
> We have also observed that the scaling law performs well in transfer learning tasks, as shown in Table 3, but does not yield the same success in STS tasks, as shown in Table 1. We think that the main reason for this discrepancy may be related to the issue of anisotropy in sentence representation [1,3], which harms the performance on STS tasks. To validate this, we measure the anisotropy of sentence embeddings across different model sizes following [4].
>
> |                    | sentence anisotropy |
> | ---------- | :--------: |
> | PromptEOL OPT-1.3b |               0.565 |
> | PromptEOL OPT-2.7b |               0.589 |
> | PromptEOL OPT-6.7b |               0.694 |
> | PromptEOL OPT-13b  |               0.719 |
> | PromptEOL OPT-30b  |               0.751 |
> | PromptEOL OPT-66b  |               0.715 |
>
> Through our analysis, we have observed that the anisotropy of sentence embeddings increases as the model size increases. While larger models may possess greater overall capacity, the issue of anisotropy may limit their performance.
>
>
> **References:**
>
> [1] Gao T, Yao X, Chen D. SimCSE: Simple Contrastive Learning of Sentence Embeddings (EMNLP 2021)
>
> [2] Yan Y, Li R, Wang S, et al. ConSERT: A Contrastive Framework for Self-Supervised Sentence Representation Transfer (ACL 2021)
>
> [3] Ethayarajh K. How Contextual are Contextualized Word Representations? Comparing the Geometry of BERT, ELMo, and GPT-2 Embeddings (EMNLP 2019)
>
> [4] Jiang T, Jiao J, Huang S, et al. PromptBERT: Improving BERT Sentence Embeddings with Prompts (EMNLP 2022)

---

> ### Author Response · Authors · 2023-11-23
> **Does our response address your concerns?**
>
> Dear reviewer dT5b,
>
> Thanks very much for your time and effort. As the discussion phase is going to end today, could you review our responses to ensure they have addressed your concerns? We would also appreciate any additional feedback you might have for improvements.
>
> Best regards,
>
> Authors of 596

---

### Meta-Review · Area_Chair_fsVP · 2023-12-10

**Metareview:**

This paper studies of leveraging recent development of large language model to produce sentence representations. Inspired by previous work of prompting BERT-like model to produce embeddings, the authors designed a new template specifically suitable for auto-regressive model. The work further design a new in-context learning approach to take the full advantage of LLM’s in-context learning capability. In addition, the authors also tried fine-tune the LLM using contrastive objectives.

The authors actively participated in the rebuttal phase. Reviewer ViaW had good discussion with the authors and agreed he would like to raise the score to 7 the system allows. The paper has an average score of 6.  The authors also addressed other concerns from the reviewers like fair comparison, reliability to other LLMs with detailed descriptions and additional experiments.

There are two remaining concerns:

1. Regarding its novelty, the reviewers agree this work is the first to demonstrate the potential of LLMs for producing sentence embeddings, and the in-context learning is new and the evaluation of models without fine-tuning demonstrates its effectiveness. However, prompting language model to generate sentence embedding has been studied in prior work for BERT-like models, and the results of PromptEOL+ICL just perform similarly to PromptBERT and PromptRoBERTa, which diminishes the value of the work. In addition, the authors claim scaling up of sentence embedding using LLMs is unexplored in the rebuttal, which is not true, as the ST5 work cited in the paper.

2. Another legitimate concern raised by reviewer ViaW during rebuttal phase is the value of the work. As the reviewer pointed out, ` If we aim for small embedding models, fully finetuning using contrastive learning (can be assisted with in-context examples) is not so expensive, which still can be a good option. But if we aim for large embedding models, the performances do not improve, which might not be cost-effective. These concerns put the proposed method in an little embarrassing position.`, the concern remains unsolved even after the rebuttal phase.

**Justification For Why Not Higher Score:**

It is a board line paper, with valid contributions of studying sentence representations from LLMs. The numbers presented in the paper are strong and likely to be cited in the future work as an baseline. However, there are concerns about its novelty and the value of the work. Given the high standard of conference, i incline to reject the paper. But can revise my decision others have strong opinion on it.

**Justification For Why Not Lower Score:**

N/A

---

### Decision · Program_Chairs · 2024-01-16

Reject